# Possible Side Effects of Polyphenols and Their Interactions with Medicines

**DOI:** 10.3390/molecules28062536

**Published:** 2023-03-10

**Authors:** Aleksandra Duda-Chodak, Tomasz Tarko

**Affiliations:** Department of Fermentation Technology and Microbiology, Faculty of Food Technology, University of Agriculture in Krakow, Al. Mickiewicza 21, 31-120 Kraków, Poland

**Keywords:** polyphenols, side effect, prooxidant activity, toxicity, mutations, DNA damage, drug interactions, cytochrome P450

## Abstract

Polyphenols are an important component of plant-derived food with a wide spectrum of beneficial effects on human health. For many years, they have aroused great interest, especially due to their antioxidant properties, which are used in the prevention and treatment of many diseases. Unfortunately, as with any chemical substance, depending on the conditions, dose, and interactions with the environment, it is possible for polyphenols to also exert harmful effects. This review presents a comprehensive current state of the knowledge on the negative impact of polyphenols on human health, describing the possible side effects of polyphenol intake, especially in the form of supplements. The review begins with a brief overview of the physiological role of polyphenols and their potential use in disease prevention, followed by the harmful effects of polyphenols which are exerted in particular situations. The individual chapters discuss the consequences of polyphenols’ ability to block iron uptake, which in some subpopulations can be harmful, as well as the possible inhibition of digestive enzymes, inhibition of intestinal microbiota, interactions of polyphenolic compounds with drugs, and impact on hormonal balance. Finally, the prooxidative activity of polyphenols as well as their mutagenic, carcinogenic, and genotoxic effects are presented. According to the authors, there is a need to raise public awareness about the possible side effects of polyphenols supplementation, especially in the case of vulnerable subpopulations.

## 1. Introduction

Polyphenols are quite important secondary metabolites of plants. Up until now there are about 8000 polyphenolic structures which have been found in plants and described [1]. Among the primary functions they perform in plants are giving colour to flowers and fruits, mainly by anthocyanins. Thanks to this, insects necessary for pollination or animals that contribute to the distribution of fruits and seeds contained in them are attracted. By giving fruits and vegetables a tart taste (astringent) polyphenols cause an unpleasant sensory perception [2]), polyphenols protect plant tissues to some extent against herbivores and—thanks to their antimicrobial activity—limit the spread of pathogens in plants [3]. Some polyphenols, such as flavonoids, are also involved in energy transfer, sex determination, regulation of growth factors, photosynthesis, and morphogenesis. They also protect plants against other abiotic stressors, such as UV radiation, cold, heat, and salinity [4]. 

Polyphenols are a large group of compounds that comprise phenolic acids, flavonoids (these include flavonols, flavanones, flavan-3-ols, flavones, anthocyanins, and isoflavones), lignans, stilbenes, and—according to some classifications—tannins and coumarins [5,6]. In a human diet, they are provided mainly by plant food. Among the richest food sources of polyphenols are seasonings and dried herbs (e.g., cloves, peppermint, anise, oregano, and rosemary), green tea, cocoa, dark-coloured berries, some seeds and nuts (e.g., flaxseed, soybean, chestnut, and hazelnut), and some vegetables (e.g., black olives, globe artichoke heads, and red chicory) [7,8]. However, various fresh, frozen, and dried fruits should also be taken into account (e.g., pomegranate and tropical fruits) [9,10,11]. In addition, many people take polyphenols in the form of dietary supplements that contain either purified polyphenols (e.g., quercetin [12] and resveratrol [13]) or extracts from raw materials which are rich in these ingredients (such as extracts from tea, red wine, grapes, blueberries, pomegranate, etc.) [11,14,15,16]. Unfortunately, even when phenolic compounds occur in the human diet in large quantities, they do not always show high biological activity immediately after consumption [5]. Their activity depends on both their bioaccessibility (the amount of an ingested nutrient that is available for absorption in the gut after digestion) and their bioavailability (the fraction of an ingested nutrient that reaches the systemic circulation and the specific sites where it can exert its biological action). The main factors which affect the bioavailability/bioaccessibility of polyphenols are food matrix, food processing, and digestive enzymes [17,18]. Moreover, some polyphenolic compounds can exert their biological activity only after the biotransformation by intestinal microbiota [6]. 

## 2. Role of Polyphenols as Antioxidants

Reactive oxygen species (ROS) play an important role in many biological processes in the human body (e.g., cellular signalling, fighting pathogens, and regulating blood pressure) [19,20,21,22]. They are naturally produced in animals’ body or plant tissue in various physiological processes, and can also be of exogenous origin [23,24,25]. The imbalance between the production and accumulation of ROS in cells and tissues and the ability of the cell/organism to deactivate them is called oxidative stress. It can be caused both by excessive ROS generation and by the impairment of antioxidant mechanisms. Usually, excessive activity of ROS can result in cells damage and the development of various diseases; therefore, it is usually considered undesirable and harmful [26,27,28,29,30,31]. However, in some cases, oxidative stress which was intentionally triggered can be used to treat cancers [32]. 

Many organisms have developed various strategies that provide antioxidant defence. These include the scavenging of free radicals, the quenching of ROS or the prevention of their generation, and the termination of radical reactions. Antioxidant defence systems comprise both enzymatic (e.g., the activity of superoxide dismutases (SOD), glutathione peroxidase (GPx), glutathione-reductase (GR), catalase (CAT), and haem oxygenase (HO)) and nonenzymatic mechanisms [28,33,34]. The The latest includes the activity of endogenous agents (e.g., glutathione (GSH), the thioredoxin system, melatonin, and coenzyme Q10) and a large group of exogenous antioxidants, which are usually provided by food, such as vitamins, polyphenols, carotenoids, and some minerals [19,35,36].

Polyphenols, with their numerous functionall groups, double bonds, and aromatic rings, have an ideal structure to act as effective antioxidants [1,6,37]. First of all, polyphenols can scavenge already-formed free radicals, such as a hydroxyl radical (^•^OH) or the superoxide anion radical (O_2_^−•^), as well as quench reactive oxygens species such as hydrogen peroxide (H_2_O_2_) or singlet oxygen (^1^O_2_) by donating a single electron (SET) or by hydrogen atom transfer (HAT) [38,39,40,41,42]. Thus, they might prevent the initiation and interruption of yet-to-be-initiated radical reactions, such as the peroxidation of lipids, the oxidation of proteins and sugars, and oxidative damage to nucleic acids [1,43]. Moreover, polyphenols are able to chelate the ions of transition metals (e.g., Fe and Cu), thus preventing the formation of free radicals in the Fenton and Haber–Weiss reactions [38,44]. Polyphenols can also function as co-antioxidants and, thus, they are involved in the regeneration of essential vitamins [1]; they are also involved in the regulation of numerous signalling pathways which are responsible, among others, for energy metabolism, adipogenesis, antioxidant, and anti-inflammatory reactions in cells [45,46]. 

Therefore, polyphenols, due to their antioxidant properties and the ability to quench free radicals and reactive oxygen species, exert a beneficial effect on human health, and they are believed to slow down the aging process as well as to be useful in the prevention of the development of many diseases [5,47,48]. 

## 3. Polyphenols in Disease Prevention and Treatment

Polyphenols, due to their antioxidant, anti-platelet, and anti-inflammatory properties, can prevent or reduce the symptoms of various metabolic diseases, such as metabolic syndrome (which includes type 2 diabetes mellitus, central and abdominal obesity, systemic hypertension, and atherogenic dyslipidaemia) [49,50,51,52,53,54,55,56], as well as cardiovascular diseases (e.g., atherosclerosis, myocardial infarction, heart failure, and stroke) [57,58,59,60,61]. Polyphenols inhibit LDL oxidation and increase HDL, improving endothelial function [62], reduce blood pressure [63], and improve the glucose homeostasis [64]. In vivo studies in rats has shown that animals which were fed on 10% hesperidin had a reduced plasma level of triglyceride as compared to control [65]. Moreover, although the ingestion of hesperidin did not exert any changes in daily food intake, body weight gain, or food efficiency, the fecal lipid content increased, suggesting inhibition of the pancreatic lipase. A significant decrease in serum lipid and lipid peroxidation was also proved in rats when galangin was administered at the dose of 50 mg/kg for six weeks [66]. The inhibitory activity of polyphenols against pancreatic lipase can be beneficial, for example, it may help with body mass reduction in obese individuals [67,68]. 

One of the most dangerous factors linked to deaths caused by cardiovascular diseases is hypertension. Polyphenols both from green and black tea significantly lower blood pressure [69,70,71]. Although the exact mechanism is not elicited, it is probable that tea polyphenols play an important role in relaxing smooth muscle contraction, enhancing endothelial nitric oxide synthase activity, reducing vascular inflammation, and inhibiting rennin activity and antivascular oxidative stress.

It was proven that a diet rich in polyphenols or supplemented with phenolic compounds also has beneficial effects in systemic and neurodegenerative diseases and amyloid diseases, particularly Alzheimer’s disease and Parkinson’s disease, as well as on cognitive functions [52,72,73,74,75,76,77,78].

Polyphenols are also important anti-cancer agents [79]. Tea polyphenol compounds were proven in many animal models to prevent various types of cancers, e.g., ovarian, breast, pancreas, colorectal, oesophagus, liver, lung, and kidney cancers [80,81,82,83,84,85,86,87,88]. Moreover, in vitro studies have demonstrated that the anticancer activity of polyphenols relies on their ability to inhibit growth or stimulate apoptosis in various cancer cell lines, such as HL60 human promyelocytic leukaemia cells, A431 epidermoid carcinoma cells, LNCaP and androgen-insensitive PC-3 human prostate carcinoma cells, human prostate cancer LNCaP cells, OVCA 433 human ovarian cancer, HepG2 cells, Caco-2 human colon cancer cells, human pancreatic carcinoma cell line Mia PACA-2, and oestrogen-independent MDA-MB-435 and oestrogen-dependent MCF-7 breast cancer cell lines [89,90,91,92,93,94,95,96,97,98]. There are also results of in vivo studies that indicate that particular classes of polyphenols, supplied with in an individual’s diet, may reduce the risk of certain cancers, e.g., colorectal [99], oesophageal [100], gastric [101], breast [102], and prostate [103], while low intake from food might increase the risk of lung cancer [104]. Polyphenols are also used to support anti-cancer therapy [105,106,107].

Among the mechanisms of anticancer activity of polyphenols are antioxidant/prooxidant activity, inhibition of specific protein kinases and other enzymes and the resulting changes in cellular signalling, inhibition of angiogenesis, estrogenic/antiestrogenic activity, antiproliferation, induction of detoxification enzymes, regulation of the host immune system, and anti-inflammatory activity [79,108,109,110]. Some polyphenols (e.g., quercetin, kaempferol, curcumin, resveratrol, and EGCG) downregulate the expression of histone deacetylases; this means that these compounds exert anticancer action by restoring epigenetic alterations in cancer cells as well as by DNA methylation and histone modifications which may prevent normal cells from turning into tumours [111]. It should be underlined that sirtuins (from SIRT1 to SIRT7) are enzymes belonging to class III HDAC (NAD+ −dependent histone deacetylases). They modulate many physiological processes in cells, such as gene expression, cellular survival, DNA repair, metabolism, energy homeostasis, stress resistance, cellular senescence and aging, inflammation-immune function, mitochondrial biogenesis, and many more [112,113,114]. It has been proven that some polyphenols, especially resveratrol, but also quercetin, naringenin, curcumin, and others, are able to activate sirtuins [115,116,117,118]. Therefore, they can regulate various physiological processes or extend the lifespan.

A very important role of diet polyphenols is to alleviate undesirable menopausal symptoms. Isoflavones, such as genistein and daidzein, are phytoestrogens which are metabolized by intestinal microbes to bioactive metabolites as O-desmethylangolensin (O-DMA) and S-equol [6]. These metabolites are oestrogen antagonists—they strongly bind to oestrogen receptors, especially ERβ, and thus exert estrogenic activities [119,120]. Soy isoflavones and their metabolites might also modulate the cytokine-induced natural killer cell function [121].

Due to their bacteriostatic or bactericidal effect, polyphenols also have a beneficial effect on the functioning of the gastrointestinal tract. On the one hand, they inhibit the growth of various pathogens, e.g., *Helicobacter pylori* [122,123,124], *Pseudomonas aeruginosa* [125,126], *Escherichia coli* [127,128,129,130], *Streptococcus mutans* [131,132], *S. aureus* [133,134,135], *Salmonella enteritidis* [136], *Vibrio cholerae* [137], *Klebsiella pneumoniae* [138], *Yersinia enterocolitica* [139], *Listeria monocytogenes* [140,141], *Candida albicans* [142], *Bacteroides fragilis, Clostridium perfringens*, and *Clostridium difficile* [143]. On the other hand, some polyphenols can stimulate the growth of beneficial bacteria. For example, polyphenols from grape enhanced the growth of *Akkermansia muciniphila* [144,145], while the growth of *Lactobacillus* and *Bifidobacterium* was stimulated by rutin [146], resveratrol [147,148], cocoa polyphenols [149], blueberry anthocyanidins [150], and apple procyanidin B2 and chlorogenic acid [151].

Considering all of the above, polyphenols seem to be our great ally, helping our bodies to function well. Therefore, diets which are rich in polyphenol compounds, as well as dietary supplements containing them, have become popular. However, it should not be forgotten that the consumption of polyphenolic compounds, especially in large amounts in a purified form (supplements instead of fruits and vegetables), can cause side effects or even have a negative impact on our health. Below is a comprehensive review of the current state of knowledge about the possible adverse effects that polyphenols can have on cells, organs, or entire organisms.

## 4. Possible Negative Consequences of Blocking Iron Uptake

Iron is an essential trace element for human life, and although it is quite abundant in the environment, iron deficiency is a common world-wide disease. A low iron level in the body can be caused both by the low intake of this metal (in some populations), and also by its low bioavailability. According to the World Health Organization (WHO), iron deficiency is responsible for ~50% of anaemia cases, and anaemia affects approximately 25% of the population in both developing and developed countries [152]. The highest anaemia prevalence has been reported in preschool-age children (47.4%) and pregnant women (41.8%). When geographical regions were analysed, the highest proportion of individuals affected were in Africa (47.5–67.6%). It should be underlined that iron deficiency can occur both when the total body iron stores are low or exhausted (absolute) and when the iron is present at normal concentration but is unavailable for the cells and body (functional) [153]. Hepcidin, which regulates the release of iron, is responsible for the regulation of iron homeostasis in the body and plays an important role in functional iron deficiency [154,155,156,157].

The bioavailability of iron from diet depends on its form. Haem iron is derived from hemoglobulin and myoglobulin from animal food sources, while non-haem iron derives from plants and iron-fortified foods, and the latter is characterized by a lower bioavailability due to much-less effective intestinal absorption [157]. 

Polyphenols are able to chelate the ions of transition metals (e.g., Fe and Cu), thus inhibiting the formation of free radicals in the Fenton and Haber–Weiss reactions. Both the binding strength and the number of bound ions depend not only on the structure of the polyphenolic compound, but also on the pH or the form of the ion (Fe^2+^ vs. Fe^3+^ and Cu^+^ vs. Cu^2+^) [44,158] (Figure 1). It is considered that this activity of polyphenols is beneficial for the body, as it will limit the formation of free radicals. Moreover, the iron-chelating activity of polyphenols is commonly used in the treatment of iron overload, which is a high risk factor for many diseases, especially human chronic diseases [159]. However, this activity can also have detrimental effects, such as in individuals with iron deficiency. A diet rich in polyphenols or polyphenols supplementation causes these compounds to bind to Fe in the intestine so that it cannot be absorbed, leading to the development of anaemia. They can also interact with the regulation of iron homeostasis.

Among potent inhibitors of iron absorption are various teas which are rich in catechins. Hurrel et al. [162] demonstrated that drinking beverages that contain 20–50 mg total polyphenols per serving caused a reduction in Fe absorption from a bread meal by 50–70% (compared to drinking water), whereas beverages containing 100–400 mg total polyphenols/serving reduced Fe absorption by 60–90%. The most potent inhibitors of Fe absorption were black teas (79–94%), peppermint tea (84%), pennyroyal (73%), and cocoa (71%), and the inhibition was reduced in a dose-dependent fashion depending on the content of total polyphenols. 

The effect of green tea extract (GTE) supplementation on the body mass, lipid profile, elements and glucose level, and antioxidant status of obese patients was monitored in 46 obese patients in a randomized, double-blind, placebo-controlled study [163]. Patients received either 379 mg of GTE or a placebo, daily for three months. The treatment with GTE allowed for mass reduction and caused a significant (*p* < 0.05) reduction in total cholesterol and triglycerides, low-density lipoprotein, and glucose, as well as in the blood Fe concentration, while the high-density lipoprotein, Zn, and Mg levels increased significantly. 

Main tea polyphenol (-)-epigallocatechin-3-gallate (EGCG) as well as rich-in-polyphenols grape seed extract (GSE, used within physiological levels, significantly decreased the transepithelial iron transport in Caco-2 intestinal cells [164]. Interestingly, the cellular level of Fe in cells was increased, although the Fe transfer across the basolateral membrane of the enterocytes was extremely low, and a higher inhibition was reported for EGCG. The authors suggest that the basolateral exit of iron via ferroportin (FPN), a transmembrane protein that transports iron from the inside of a cell to the outside (see Figure 2), was impaired by the formation of a non-transportable complex Fe-polyphenol. Similar results were obtained by Ma et al. [165], who reported that EGCG and GSE decreased the transepithelial transport of haem-derived iron by Caco-2 cells, mainly by the reduction of basolateral iron exit. In another study, the impact of EGCG, GSE, and GTE on haem iron (heme-^55^Fe) absorption by Caco-2 cells was analysed [166]. The tested polyphenols significantly inhibited heme-^55^Fe absorption in a dose-dependent manner. However, the addition of ascorbic acid was able to balance or even reverse the inhibitory effects of polyphenols when they were used at lower concentrations (≤4.6 mg/L). 

Coherent results were obtained in a study on rats and Caco-2 cells [167]. Quercetin (Q), in the form of aglycone or as methylated (M )derivatives 3-MQ, 4′-MQ, and 3,4′-diMQ, were introduced into rat duodenum together with radioactive iron ^59^Fe. An increase in the mucosal uptake of ^59^Fe was observed (especially for Q and 4′-MQ) in comparison with the untreated control group. On the other hand, the release of ^59^Fe from the intestinal mucosa into the blood was significantly diminished in the presence of Q and 4′-MQ in comparison with 3-MQ, 3,4′-diMQ, and the untreated controls. Both the stimulation of Fe uptake and the inhibition of Fe efflux in the presence of Q were also confirmed in Caco-2 cell monolayers. As the methylation of the 3-OH negated both the increase in apical iron uptake and the inhibition of basolateral iron release, the authors postulated that the mechanism is dependent on the iron chelation between 3-hydroxyl and 4-carbonyl groups of quercetin. Probably, the quercetin-iron complex is too large to exit enterocytes via ferroportin (FPN), although some efflux was still possible through the glucose transporters. Moreover, the authors observed that, in the long term, the decrease in iron transport across Caco-2 cells caused by Q was associated with a significant, dose-dependent decrease in FPN protein and FPN mRNA in Caco-2 cells, and it was mediated by the interaction of miRNA with the 3′UTR of FPN mRNA.

Quercetin can decrease the intracellular labile iron pool by binding labile iron and transferring it from the cell compartment to the transferrin molecule [170]. The complexes of quercetin and rutin with iron were proven to permeate cell membranes; however, only free quercetin was able to access to the cytosol and remove iron from cells.

The interaction of quercetin and the related flavonoids with iron supplements in an animal model of iron deficiency anaemia (IDA) was assessed by Mazhar et al. [171]. In the study, female weanling Sprague Dawley rats were kept on a low iron diet for 20 days to induce IDA (manifested by a reduced level of haemoglobin, haematocrit, and serum iron). Then, the animals obtained a dose of 50 mg/kg of ferrous sulphate (FeSO_4_) supplement combined in an equal ratio with quercetin (Q), quercetagetin (QTG), and patuletin (PAT) for 30 days. The orally-administered doses of a combination of flavonoids and FeSO_4_ which were given to the rats significantly improved both the serum levels and spleen tissue availability of iron compared to the IDA model animals. The authors postulated that a complex of iron with quercetin might provide an alternative pathway for iron absorption through the glucose transporter. Moreover, they observed a slightly lower FPN expression in the spleens of rats with the Q-FeSO_4_ combination than for FeSO_4_ alone; however, QTG-FeSO_4_ and PAT-FeSO_4_ increased the FPN. The different impacts of polyphenols were probably caused by the differences in the functional group at the C6 position of their molecules.

The impact of quercetin was also evaluated in a randomized, double-blind, placebo-controlled study performed in 90 patients with non-alcoholic fatty liver disease. The patients were supplemented with either a quercetin or a placebo capsule twice daily (500 mg) for 12 weeks and their blood parameters were assessed. The results showed that ferritin was significantly (*p* = 0.013) decreased, while whole red blood cells (RBC) increased compared to the placebo group [172]. According to the authors, two mechanisms may be involved. First, quercetin acted as an electron donor in enterocytes and, thus, might facilitate inorganic iron absorption; however—simultaneously—it could reduce the expression of FPN and hephaestin (both responsible for iron transfer from the enterocyte to the blood, see Figure 2). Another possibility is that the quercetin–iron complex is retained in the enterocyte cytosol, which results in a functional iron deficiency and implicates a decrease in ferritin. 

Inconsistent results were obtained in a study by Al-aboud et al. [173]. They showed that consuming 8 g of raisins (*Vitis vinifera* L.) for 20 days resulted in increased haemoglobin and serum iron levels, an increase in ferritin, and a decrease in total iron binding capacity (TIBC) and transferrin. However, the group of volunteers was small (seven women) and raisins, besides various polyphenols, also provide iron and vitamin C. It is known that ascorbic acid enhances the absorption of non-haem iron, mainly due to its ability to reduce Fe^3+^ to Fe^2+^, which makes it available for transport by DMT1 (Figure 2). Therefore, the intake of food that is rich in vitamin C should be critical when a given diet is abundant in the inhibitors of iron absorption (polyphenols, phytate, calcium, and proteins) [161].

To sum up, the influence of polyphenols on the level of iron requires a more thorough analysis, because both its excess (causing, e.g., excessive ROS production) and its deficiency (anaemia) are harmful. The oral iron therapy of IPA often causes gastrointestinal side effects such as constipation, nausea, diarrhoea, abdominal pain, vomiting, heartburn, dark stools, and flatulence [174]. Moreover, an excess of unabsorbed iron can lead to dysbiosis by modifying the microbiota profile because iron favours the growth of some pathogenic bacteria (e.g., *Escherichia coli*, *Shigella*, *Campylobacter*, *Salmonella*, *Clostridium*, and *Bacteroides)* and also lowers the abundance of beneficial species belonging to *Lactobacillus* and *Bifidobacterium* [175]. Some of pathogenic species of intestinal bacteria can produce genotoxic metabolites which promote inflammation and carcinogenesis. Therefore, further studies are also important from the point of view of the possibility of using polyphenols’ ability to bind iron in cancer therapy [176]. The ability of flavonoids to chelate iron can be used in the production of more effective drugs or supplements for IDA treatment and prevention [174]. 

## 5. The Inhibition of Digestive Enzymes by Polyphenolic Compounds

Flavonoids are able to form complexes with proteins, both through nonspecific forces (e.g., hydrogen bonding and hydrophobic effects) and by covalent bond formation [177]. As a result of protein binding by polyphenols, they are sequestered into soluble or insoluble complexes, which affects the proteins’ function, structure, solubility, hydrophobicity, thermal stability, and isoelectric point, as well as their susceptibility to digestive enzymes [178,179]. These changes can be crucial for the functioning of the whole organism, as the digestibility and utilization of food proteins can be affected. Moreover, the activity of digestive enzymes (i.e., amylases, proteases, and lipases), which are also proteins, can be affected by binding with polyphenols, and the changes in the enzyme structure implicate an impaired function and disturbances in the course of biochemical reactions or processes that a given enzyme catalyses.

Of course, in some diseases, the ability to inhibit certain enzymes may be beneficial, e.g., in the treatment of diabetes or to improve postprandial blood glucose [180,181], as well as in body mass reduction or reducing fat absorption in obese people. However, there are some subpopulations where this generally beneficial effect may have an undesirable effect. Also, in healthy individuals, any disorders in the activity of digestive enzymes are unfavourable, as they cause unpleasant symptoms in the digestive system, as well as the impaired assimilation of some nutrients, which in turn may adversely affect the activity of the organism. The efficient functioning of digestive enzymes, especially glucose-releasing enzymes, is particularly important in people practicing endurance sports, and research has shown that these athletes quite often struggle with gastrointestinal disorders. Along with an increasing intensity of exercise, the blood flow (and its oxygen, nutrients, and ability to clear waste products) naturally changes, and the blood is moved away from the stomach and intestines and directed toward the working muscles to facilitate the movement [182]. This reduced blood flow in the gut can cause side effects, especially if the athlete, during a very intense exercise-to-exhaustion or long duration/endurance exercise, is feeding with fluids and food while minimal digestion is occurring. As a consequence, the gastrointestinal tract can become more permeable. In such a situation, any additional disturbance in the work of digestive enzymes (e.g., inhibition of enzyme activity caused by the polyphenols) can have harmful effects on the body.

Endurance athletes or long-distance runners are known to often experience digestive problems [183]. Among the known symptoms of digestive enzymes deficiency are food intolerances, poor nutrient absorption, a weaker immune system, weight gain and obesity, skin and digestive problems, etc. Moreover, long-term endurance training for a long duration causes changes in the activity of digestive enzymes (e.g., pancreatic amylase) [184]. The supplementation of digestive enzymes by athletes is, apart from a method to alleviate unpleasant symptoms, one of the methods of improving sports performance. There are several studies showing that enzyme supplementation can help prevent indigestion and improve nutrient absorption. For example, men who consumed whey protein concentrate (WPC) with patented digestive proteases (Aminogen^®^) had an increased rate of absorption of processed WPC (compared to the controls) [185]. As well, statistically significant increases in amino acids level were reported, both the total serum amino acid level and the level of individual serum amino acids. Significant decreases in CRP level were also reported. In another study, protease supplementation (two tablets with 325 mg pancreatic enzymes, 75 mg trypsin, 50 mg papain, 50 mg bromelain, 10 mg amylase, 10 mg lipase, 10 mg lysozyme, and 2 mg chymotrypsin) attenuated muscle soreness after downhill running, facilitated muscle healing, and allowed for faster restoration of contractile function after intense exercise [186]. Therefore, it is worth making athletes aware that simultaneous supplementation with enzymes and preparations rich in purified polyphenols may offset the beneficial effects of these supplements.

It was found that dietary polyphenols can decrease glucose uptake, as was confirmed in Caco-2 cells, a cell line which is used as a model of the intestinal epithelial barrier [187]. The authors investigated the effect of various polyphenols on both the active intestinal transport (sodium-dependent) and facilitated transport (sodium-free) of glucose. The results suggest that aglycones (apigenin, myricetin, phloretin, and quercetin) could inhibit facilitated glucose uptake, whereas glycosides (such as naringin, phloridzin, rutin, and arbutin) inhibited the active transport of glucose. The non-glycosylated dietary polyphenols (such as (+)-catechin, (−)-epicatechin, (−)-epigallocatechin (EGC), (−)-epicatechin gallate (ECG), and (−)-epigallocatechin gallate (EGCG)) appeared to exert their effects via steric hindrance, and EGCG, ECG, and EGC were effective against both transporters. Phenolic acids did not have any effect.

Thus, it can be seen that the proper functioning of digestive enzymes is important in healthy people, and any disturbance in their operation may translate into, for example, a worse performance in sports.

Another group for which the inhibitory effect of polyphenols on digestive enzymes may be harmful is people with food intolerances resulting from the deficiency or lack of certain enzymes. The most common digestive disorders of this type include gluten intolerance, celiac disease, lactose intolerance or complex carbohydrate intolerance, exocrine pancreatic insufficiency, cystic fibrosis, and pancreas cancer. Some of these diseases are genetic, while others may increase with age [188,189]. It is estimated that adverse reactions to food, including food intolerances and food allergies, may affect up to 20% of the general population [190], whereas, among patients with irritable bowel syndrome (IBS), the incidence of food intolerance may reach 65% [191]. Therefore, supplementation with digestive enzymes constitutes an increasingly common method of supporting the body in the digestion of proteins, carbohydrates, and lipids in people suffering from various digestive disorders, from lactose intolerance to celiac disease to cystic fibrosis [192,193,194,195,196,197]. In all of these cases, knowledge of food ingredients that may additionally inhibit the already-low activity of specific enzymes is essential for proper diet planning and meal composition in such patients to prevent malnutrition.

The last large group affected by a reduced activity of digestive enzymes is elderly people. Some studies conducted in humans showed that individuals above 65 years old had significantly reduced (compared to young controls) bicarbonate and enzyme (lipase, chymotrypsin, and amylase) secretions, due to both a decrease in secreted volume and enzyme concentrations [198,199,200,201,202]. Moreover, the biodiversity and stability of the intestinal microbiota are reduced in the elderly, which significantly correlates with frailty, co-morbidity, nutritional status, and markers of inflammations. An age-related decrease in digestive enzymes and changes in microbiota might lead to a reduced availability and absorption of some nutrients, such as carbohydrates, lipids, amino acids, minerals (calcium and iron), and vitamins (B12, B6, folic acid, and lipid soluble vitamins), which in turn increases the risks for the development of a range of pathologies associated with most systems, in particular the muscoskeletal-, nervous-, cardiovascular-, immune-, and skin systems [203,204,205]. Various clinical trials have proven that enzyme supplementation has great potential in the treatment of diseases caused by a deficiency of various enzymes [192,193,206,207]. However, the right composition of diet is also of crucial significance because various studies have demonstrated that naturally occurring polyphenols, e.g., condensed tannins, can inhibit a number of digestive enzymes, including α-amylase, α-glycosidase, pepsin, trypsin, lipase, and chymotrypsin. Therefore, they influence the digestion process and can further diminish the availability of nutrients. 

Table 1 summarizes the known and scientifically documented interactions between dietary polyphenols and enzymes that lead to a reduction or inhibition in the activity of digestive enzymes. It should be emphasized, however, that most of the known studies have been conducted in vitro. In vivo studies in humans or animal models are rare and the results are often inconclusive. For example, quercetin was proven to be a promising pancreatic lipase inhibitor by reducing fat absorption in rats in vivo [208]. Pre-administration with 5 and 10 mg quercetin per kg of body weight significantly reduced fat absorption and, correspondingly, significantly increased fat excretion in rat faeces. The inhibition of quercetin on lipase can last at least 2 h in vivo.

As was presented in the Table 1, there are large differences in the inhibitory activity of various polyphenols. Therefore, some meals containing plant-raw material can exert stronger inhibitory activity than other apparently similar plants. Because of this, the patient, in consultation with a doctor or nutritionist, can choose the right fruits and vegetables or dietary supplements to support their digestive processes, instead of harming them. One example is some soft fruits; strawberry and raspberry can inhibit α-amylase more than blueberry, blackcurrant, or red cabbage, whereas α-glucosidase was more readily inhibited by blueberry and blackcurrant [238]. Among various berries, strawberry and raspberry, as well as arctic bramble and cloudberry, were proven to have a high potential to inhibit lipase activity, even at low levels, which are achievable in the gut after the intake of a small amount of berries [239]. On the other hand, extract from rowan berries strongly inhibited α-amylase (IC_50_ 4.5 µg GAE/mL), while polyphenol-rich extracts from blackcurrants inhibited α-glucosidase activity (IC_50_ 20 µg GAE/mL). 

It is worth noting that some polyphenols or polyphenol-rich products delay gastric emptying and lower the postprandial glucose response; therefore, they alter the apparent glycaemic index of food [55,240,241,242,243], which is a rather beneficial influence. However, in particular subpopulations (e.g., people with functional dyspepsia or with eating disorders such as anorexia nervosa), delayed gastric motility would enhance undesired symptoms such as nausea, vomiting, and gastric fullness, and may lead to difficulties during refeeding and weight restoration [244,245].

## 6. Possibility of the Intestinal Microbiota Inhibition and Consequences

The negative impact of polyphenols on the functioning of the digestive system could result not only from the inhibition of digestive enzymes, but also from their influencing the intestinal microbiota. 

In general, the interactions between polyphenolic food ingredients and bacteria residing in the digestive tract are very complex. Firstly, bacterial enzymes are responsible for the release of polyphenols from the food matrix, and thus increase their bioaccessibility and bioavailability. Moreover, bacteria carry out the conversion and degradation of polyphenols, usually through hydrolysis reactions and double bond reduction reactions. The intestinal microbiota are equipped with a huge set of various enzymes that catalyse reactions which are inaccessible to human enzymes, thanks to which all ingredients that are sent to the intestine with food can be modified and used [246,247]. Intestinal bacteria are able to hydrolyse glycosides, glucuronides, ester, sulphates, amides, and lactones through the action of enzymes such as β-glucosidase, α-rhamnosidase, β-glucuronidase, α-galactosidase, esterases, and sulfatases. Moreover, they catalyse the reactions of aromatic rings cleavage, decarboxylation (decarboxylase), demethylation (demethylase), isomerization (isomerase), reductions (reductases and hydrogenases), and dehydroxylation (dehydroxylase) [248,249]. These reactions lead to the degradation of glycosides (glucosides, galactosides, rhamnosides, rutinosides, etc.) or polymeric polyphenols (such as ellagitannins, tannins, and procyanidins) to simpler compounds, i.e., aglycones, and then to phenolic acids and products of their metabolism. The bioactive properties of polyphenol metabolites are completely different from the activity of the parent compounds. Usually, metabolites have weaker antioxidant properties; however, it happens that polyphenol gains its biological activity only after bacterial biotransformation. The best-known cases are transformations of soya isoflavone daidzein to S-equol and O-desmethylangolensin, ellagitannins to urolithin A, and secoisolariciresinol to entererodiol and enterolactone [6,249]. 

On the other hand, it has been known for many years that polyphenols and foods that are rich in them (such as herbs, spices, fruits, and vegetables) have an inhibitory effect on numerous species of bacteria, both Gram-positive and Gram-negative bacteria [6]. Unfortunately, this inhibitory effect is not limited to pathogens.

Strong antimicrobial activity against *Salmonella enterica* ser. Typhimurium, *E. coli* as well as against *Lactobacillus rhamnosus* and *L. rhamnosus* GG was reported for extracts of cloudberry, raspberry, and strawberry at higher concentrations [250]. Strawberry extract was an effective inhibitor against *E. faecalis* and *Bifidobacterium lactis*. Myricetin inhibited the growth of all lactic acid bacteria derived from the human gastrointestinal tract flora; however, it had no impact on *Salmonella* Typhimurium and *Lactobacillus plantarum* from beer. A bacteriostatic impact of luteolin was reported on some *Lactobacillus* species, *Bifidobacterium lactis* and *Enterococcus faecalis*, while no impact was observed on Gram-negative bacteria. Apigenin, (+)-catechin, kaempferol, isoquercitrin, and rutin had no effect, while phenolic acids inhibited the growth of *E. coli* and *S. enterica*.

Aqueous extract of *Salvadora persica* L. inhibited, in a dose-dependent manner, all tested microorganisms, especially *Streptococcus mutans*, *S. faecalis*, and *S. pyogenes*, as well as *Staphylococcus aureus*, *Lactobacillus acidophilus*, *Pseudomonas aeruginosa*, and *Candida albicans*; however, methanol extract from the same plant was inactive against *L. acidophilus* and *P. aeruginosa* [251]. 

Vattem et al. [123] investigated the impact of 53 constituents of peppermint oil and green tea polyphenols (e.g., (−)-epigallocatechin and (−)-epigallocatechin-3-gallate (EGCG)) on various strains of *Escherichia coli.* All of the compounds had strong antibacterial activity against non-pathogenic *E. coli*, while peppermint oil, menthol, menthone, and neomenthol also killed the enterohemorrhagic strain *E. coli* O157:H7 at concentrations 400 µg/mL within 1 h. GTP inhibited the growth of pathogenic *E. coli,* although at concentration of 800 µg/mL within 18 h.

Other results were obtained by Tzounis et al. [252], who demonstrated that (+)-catechin caused a significant decrease in the growth of the *Clostridium histolyticum* group, and a marked increase in the growth of the beneficial bacterial group of *C. coccoides–Eubacterium rectale*, *Lactobacillus* spp., and *Bifidobacterium* spp. Furthermore, (−)-Epicatechin caused a significant increase in the growth of the *Eubacterium rectale–C. coccoides*.

Differences in the effects of polyphenols, depending on their form (aglycones and glycosides), on bacteria that are representative of the physiological intestinal microbiota were investigated by Duda-Chodak [146]. Quercetin (aglycone) had a strong inhibitory impact (minimal inhibitory concentration, MIC = 20–50 µg/mL) on *Ruminococcus gauvreauii, Bacteroides galacturonicus*, and *Lactobacillus* sp. growth, while rutin (glycoside) had no effect. Similarly, naringenin and hesperetin (aglycones) inhibited the growth of almost all analysed bacteria (with MIC value ≥ 250 µg/mL), whereas their glycosides, naringin and hesperidin, had no impact. 

The concentrations of individual polyphenols used in the above-described in vitro studies are possible to obtain in the human digestive tract when high concentrations of pure polyphenols are taken in the form of dietary supplements. By intaking a balanced diet, even rich in fruits and vegetables, the risk of reaching these unfavourable concentrations of polyphenols is much lower.

The inhibition of the growth of lactic acid bacteria/LAB/by 13 phenolic acids (benzoic acid, 3-hydroxybenzoic acid, 4-hydroxybenzoic acid, 4-hydroxy-3-methoxybenzoic acid, 3,4-dihydroxybenzoic acid, phenylpropionic acid, 3-hydroxyphenylpropionic acid, 4-hydroxyphenylpropionic acid, 3,4-dihydroxyphenylpropionic acids, phenylacetic acid, 3-hydroxyphenylacetic acid, 4-hydroxyphenylacetic acid, and 3,4-dihydroxyphenylacetic acid) has been reported [253]. *Lactobacillus paraplantarum* LCH7 was the most sensitive strain, while *Lactobacillus fermentum* LPH1 was more resistant. The most active compounds were: 4-hydroxybenzoic acid for *L. fermentum* CECT 5716, *L. fermentum* LPH1, *L. brevis* LCH23, and *L. plantarum* LCH17; 4-hydroxybenzoic acid and phenylpropionic acid for *L. paraplantarum* LCH7 and *L. coryniformis* CECT 5711; 3-hydroxyphenylpropionic acid for *L. fermentum* CECT 5716.

A flavan-3-ol-enriched grape seed extract inhibited the growth of both pathogens and beneficial bacteria (*Streptococcus thermophilus*, *Bifidobacterium lactis* BB12, *Lactobacillus fermentum*, *L. acidophilus*, and *L. vaginalis*), and simultaneously stimulated the growth of some *Lactobacillus plantarum*, *L. casei*, and *L. bulgaricus* strains, while it had no impact on *Bifidobacterium breve* 26M2 and *B. bifidum* HDD541 [254]. Thirteen tested polyphenols influenced the growth of beneficial bacteria, exerting both stimulatory and inhibitory impacts. Coumaric acid had a strong stimulatory effect on *Bifidobacterium bifidum*, while vanillic and caffeic acid stimulated *Bifidobacterium adolescentis*. Hesperidin and quercetin exerted an inhibitory dose-dependent impact on both of the tested bacteria [255]. In another study, the growth of *Lactobacillus* ssp. and *Bifidobacterium* spp. was inhibited by caffeic acid, 3-phenylpropionic acid, and, to a lesser extent, by gallic acid, while *Lactobacillus casei* Shirota growth was inhibited only by 3-phenylpropionic acid [256].

On the other hand, there are some studies showing that some polyphenols can stimulate the growth of beneficial bacteria [144,145,147,148,149,150,257,258,259]. Usually, a positive influence of polyphenols is observed for plant extracts or natural products (such as berries, cocoa, grape polyphenols, red wine polyphenols, berries extract, tannin-rich products, and pomegranate extract) that are composed of various polyphenols in their more “natural” form (e.g., glycosides), while a negative impact is observed in particular for purified polyphenolic compounds, mainly aglycones [249,260], a form that is rarely found in food but is a common form in dietary supplements.

As was mentioned above, the impact of polyphenols as well as the mechanism of this action depends both on the type of polyphenols and bacteria species. Among the mechanisms of antibacterial activity of polyphenols, the most frequently mentioned are: (i) polyphenols’ reactions with proteins; (ii) substrate and metal deprivation; (iii) the inhibition of nucleic acid synthesis by bacterial cells or DNA damage; (iv) interaction with the bacterial cell wall or inhibition of cell wall formation; (v) changes in the function of the cytoplasmic membrane (e.g., modifications of the membrane permeability or fluidity, cytoplasmic membrane damage, and—as a result—the membrane’s disruption); (vi) inhibition of energy metabolism; (vii) changes in cell attachment as well as inhibition of biofilm formation [6].

Polyphenols modulate the profile of microbiota, both by stimulating some species and by inhibiting others, which results in a changed ratio between the most important groups of bacteria—the condition called dysbiosis. Dysbiosis involves the loss of beneficial bacteria and/or an expansion of pathogenic microbes (pathobionts), which leads to an imbalance between the number of particular bacteria or to reduced microbial diversity. It has been shown that dysbiosis promotes inflammation and immunological dysregulation, and is associated with various diseases such as irritable bowel syndrome (IBS), functional dyspepsia, metabolic disorders (diabetes, obesity, and non-alcoholic fatty liver disease), colorectal cancer, inflammatory bowel diseases (Crohn’s disease and ulcerative colitis), cardiovascular diseases, and small intestinal bacterial overgrowth (SIBO) [261,262,263,264,265,266]. Instability of the gut microbiome can be caused by many factors, including infection, exercise, sleep pattern, exposure to antibiotics, various co-morbidities, and—as proven above—by diet. This, of course, does not mean that polyphenols are responsible for the mentioned diseases; however, it is possible that they can interfere with the severity of symptoms or the treatment. Recently, there is growing interest in the other serious consequences of gut microbiota imbalance. Gut dysbiosis impairs the function of the gut–brain axis, which is associated with the development of various neurological and psychiatric disorders such as anxiety, stress, major depressive disorder, schizophrenia, bipolar disorder, autism spectrum disorder, and attention-deficit hyperactivity disorder, as well as Parkinson’s disease, Alzheimer’s disease, dementia, multiple sclerosis, and epilepsy [267,268,269,270,271].

In conclusion, the inhibition of intestinal microbiota by polyphenols implicates changes in their physiological role, e.g., in the amount of all bacterial metabolites that are important for human health, such as vitamins, amino acids, SCFA, antimicrobial peptides, and neurotransmitters [265,272]. Moreover, the participation of microbiota in detoxication is also disturbed, as these microorganisms are responsible for the degradation not only of food ingredients, but also of a number of xenobiotics (such as drugs), in particular, compounds with branched chains or aromatic rings [272,273,274].

## 7. Interactions of Polyphenolic Compounds with Drug Disposition and Metabolism

Polyphenols, due to numerous active functional groups, interact not only with reactive oxygen species or free radicals, but also with other chemical compounds that they “encounter” in their environment. Among such interactions, it is worth mentioning some that can have a very negative impact on human health. The most important are the interactions of polyphenols with the components of various drugs (e.g., with iron-containing preparations used to treat anaemia) and the influence of polyphenols on drug metabolism, as well as pharmacokinetics, which may result both in increasing and diminishing their therapeutic effect. In general, consumers are often aware of such interactions, and, for example, they know that medicines should not be taken with grapefruit juice or herbal infusions. However, they are not fully aware of what lies at the basis of such recommendations. The basic mechanism relies on the impact of polyphenols on the activity of drug-metabolizing enzymes, such as phase I and phase II enzymes, e.g., cytochrome P450, glutathione S-transferase, UDP-glucuronosyltransferase, sulfotransferase, N-acetyltransferase, methyltransferase, epoxide hydrolase, and NAD(P)H:quinone oxidoreductase [109,275,276,277,278,279,280,281,282].

The cytochrome P450 (P450 or CYP) is a group of haem-containing isoenzymes that are responsible for the metabolism of a wide range of endogenous compounds (steroid hormones, lipids, and bile acids), as well as various xenobiotics, mainly those of hydrophobic nature, including drugs, carcinogens, environmental pollutants, and dietary products [283,284]. It is also the final element of the electron transport chain, in which electrons are transferred to an oxygen atom in the O_2_ molecule, reducing it to the H_2_O molecule [285]. The basic P450 reactions catalysed by P450 include C- and N-hydroxylation; N-, O-, and S-dealkylation; N- and S-oxidation; epoxidation; dehalogenation; ring contraction and formation; dehydration; C-C bond cleavage; isomerization; reduction; oxidative deamination [280,284]. 

Various isoenzymes occur widely in almost all tissues; however, they show the greatest activity in the liver and gut [280]. In the human genome, there are above one hundred different genes (with names that begin with CYP) that code for different cytochromes P450 and the number following the letters “CYP” indicates the gene family, while subfamilies are represented by a letter that is followed by yet another number to indicate the specific gene (e.g., for the enzyme CYP3A4, “3” stands for the gene family, “A” for the subfamily, and “4” defines the gene that encodes a specific polypeptide) [286]. Cytochromes that belong to families 1, 2, and 3 are the principal xenobiotic metabolizers, while the others are involved in the biotransformation and elimination of various endogenous biomolecules such as fatty acids and hormones. The most significant CYP isoenzymes and the most abundant in humans are CYP3A4 and CYP2D6, which are present mainly in the liver and the gut wall [280,285,287].

Polyphenols, as well as polyphenol-rich food (e.g., herbs, spices, and fruit), can alter drug absorption, distribution, and metabolism, the latter both by the inhibition of P450 activity (the basic types of enzyme inhibition are competitive, non-competitive, and uncompetitive), and the reduction of P450 activity, which of course directly influences the clinical effects of drugs [110,281,282,288,289,290,291,292,293]. When the metabolism of a drug is limited, its concentration in the blood or tissues increases, causing various effects which are sometimes very dangerous. On the other hand, the induction of P450 activity diminishes the duration of action of a drug by increasing its metabolic elimination, which is also an undesirable effect and may pose a serious risk [109]. In addition, polyphenols can also affect drug transport through their interaction with the drug transporters, e.g., P-glycoprotein belonging to ABC transporters [294], organic anion transporting polypeptides (OATPs), and organic cation transporters (OCTs) [293,295].

This means that there is a considerable risk for an adverse impact of the drug–polyphenol interactions, especially for drugs with a narrow therapeutic index such as warfarin, cyclosporine A, and digoxin. To avoid these side effects, both the patient and the doctor should be aware of the known interactions between the most commonly used drug and various kinds of food or herbal preparations. Known interactions are summarized and presented in Table 2.

Summarizing the available literature, the most important polyphenols that exert inhibitory activity against drug-metabolizing enzymes include quercetin and its derivatives [331,332], resveratrol [282], chrysin [333], anthocyanins, naringenin [334,335], apigenin [336], coumarins [337], kaempferol [332], acacetin [338], luteolin, diosmetin [339], caffeic acid [340], and gallic acid [109,341]. There are also some “dangerous” plants that, if consumed along with medications that are permanently taken, should be carefully controlled, and the concomitant consumption should be under the supervision of a physician. These include, among others, grapefruit, orange, grape, apple, goji berries, raspberry, cranberry, mulberry, mango, tangerine, ginger, green tea, St. John’s Wort, avocado, spinach, and tomato juice [295,342]. Therefore, both physicians and other health care providers, including pharmacists in community pharmacies, should draw the attention of patients to the fact that “natural” is not always “safe.” It is necessary that patients that permanently take medicines and want to use herbal supplements should purchase them from a pharmacy only after pharmacist counselling. There is a strong need for proper communication about possible side effects of the concomitant use of many natural products with drugs. This means that the physician, when prescribing a drug from the group susceptible to interactions (especially, but not only, those listed in Table 2), should inform the patient in detail about potential interactions with food ingredients and explain the resulting health and even life risks [343].

## 8. Can Polyphenols Induce a Hormonal Imbalance?

Among polyphenols are isoflavones, which have gained popularity as an alternative treatment for menopausal symptoms for women who cannot take hormones. Isoflavones are structurally similar to oestrogens in that, in various tissues, they can exert both estrogenic and antiestrogenic properties [344]. It is very important to remember that the biological activity of isoflavonoids depends on the gut microflora composition, because only enzymes of particular bacteria can conduct the conversion of isoflavonoids, i.e., daidzein can be converted into its biologically active metabolites S-equol or O-desmethylangolensin, while genistein can be converted to p-ethyl phenol. Although isoflavones seem to have a positive impact on post-menopausal women, their impact on women at the reproductive age might be less beneficial. Chandrareddy [345] described the case of three women with endometrial pathology whose abnormal uterine bleeding was related to a high intake of soy products. The first of the patients had postmenopausal bleeding with uterine polyp, proliferative endometrium, and a growing leiomyoma, the second had severe dysmenorrhea, abnormal uterine bleeding, endometriosis, and uterine leiomyoma not responding to treatment, and the last woman—severe dysmenorrhea, abnormal uterine bleeding, endometriosis, and uterine leiomyomata presented with secondary infertility. In all of these cases, the problems were reduced or disappeared when soybean and soybean products were excluded from the diet.

The side effect of soy isoflavones was also reported in a placebo-controlled crossover trial conducted by Hutchins et al. [55]. The only participant, upon entry into the study, was a healthy, well-nourished, and normotensive postmenopausal woman (51 years old). She consumed the first of four randomly assigned treatments (500 mg vitamin C plus 5 mg/kg body weight soy isoflavones) and, during this treatment, her systolic blood pressure spiked to a recorded 226/117 mmHg, which required a medical intervention and discontinuation of study participation. A possible mechanism for this reaction can involve the inhibition of monoamine oxidase by the isoflavones or their metabolites (S-equol) [346,347] as well as an imbalance in the renin-angiotensin system, an important regulator of blood pressure. Angiotensinogen production by the liver can be modulated by oestrogens [348]; therefore, an increase in serum isoflavone concentrations, due to the high isoflavone intake, might stimulate an oestrogenic response, thereby increasing hepatic angiotensinogen production and its release into the plasma. This means that, regardless of the mechanism, patients who are consuming soy isoflavone supplements should be informed that elevated blood pressure may be a potential side-effect to consider and it should be monitored.

There is also evidence that, while in women the estrogenic activity of isoflavones is harmless or even beneficial, in the case of men it can lead to troubles with sexual activities. It was proven that the plasma testosterone and androstenedione levels were significantly lower in adult Sprague–Dawley rats that were fed a phytoestrogen-rich diet (containing approximately 600 µg/g isoflavones) compared with animals fed a phytoestrogen-free diet [349]. After five weeks of consuming these diets, plasma phytoestrogen levels were 35 times higher, while body and prostate weights were significantly decreased in animals fed the phytoestrogen-rich diet vs. the phytoestrogen-free fed animals. There is also a known case of a 60-year-old man with bilateral gynecomastia of six months’ duration, who also reported erectile dysfunction and decreased libido [350]. In his medical history, no changes in testicular size, no testicular trauma, no sexually transmitted diseases, no headaches, no visual changes, and no change in muscular mass or strength were reported, and testicular ultrasonography was normal. However, a laboratory assessment showed that his estrone and oestradiol concentrations were increased four-fold above the upper limit of the reference range. After another detailed interview, it was revealed that the patient drank three quarts of soy milk daily. After he discontinued drinking soy milk, his breast tenderness resolved, and his oestradiol concentration slowly returned to normal. With the exception of the case which was just described, findings from various human studies and meta-analyses show that neither isoflavone supplements nor soy products affect male reproductive hormones, and serum testosterone or oestrogen levels in men are not altered [351]. In other words, isoflavones do not exert feminizing effects on men when consumed at levels equal to and even considerably higher than are typical for males [352].

Therefore, can isoflavones actually cause hormonal imbalances? Generally, it seems that, in healthy people, isoflavones do not pose a threat; however, in the case of people with complete deficiencies or diseases, the situation may be different. Several studies have found that soy isoflavones can inhibit the thyroid hormones in people with iodine deficiency. Genistein has been shown to reduce thyroid peroxidase (TPO) (in vivo studies with rats) by up to 80% in a dose-dependent manner [353]. In another study, genistein and daidzein were proven to induce microfollicular changes in the thyroid tissue in rats and reduced the level of thyroid hormones in Orx middle-aged male rats (a model of andropause). This reduction consequently led to a feedback stimulation of pituitary TSH cells, and the stimulatory effect was higher in the daidzein-treated rats [354]. Some in vitro and in vivo studies have shown that various polyphenols can act as potent inhibitors of TPO, a key enzyme in thyroid hormone synthesis. For example, flavonoids such as fisetin, kaempferol, naringenin, and quercetin inhibited TPO with IC_50_ values ranging from 0.6 to 41 µM, while the IC_50_ values for inhibition of TPO-catalysed reactions by genistein and daidzein were ~1–10 µM [355,356]. In subjects with subclinical hypothyroidism, diet supplementation with 16 mg soy phytoestrogens caused a three-fold increased risk of developing overt hypothyroidism; a soy phytoestrogen supplementation significantly reduced the insulin resistance, hsCRP, and both systolic and diastolic blood pressure in these patients [357].

On the other hand, a randomized controlled trial among Chinese postmenopausal women that were “equol-producers” demonstrated that the consumption of both soy and purified daidzein are safe and have no detrimental effect on thyroid function [358]. Similarly, when 14 trials (thyroid function was not the primary health outcome in them) were analysed for the effects of soy products or isoflavones on thyroid function, either no effects or only very modest changes were noted [359]. Thus, the findings provide little evidence that, in euthyroid or iodine-replete individuals, soy products or isoflavones adversely affected thyroid function. This is contrary to the hypothyroid patients, in which soy foods, by inhibiting absorption, could increase the dose of thyroid hormones.

Genistein, glycitein, and daidzein were proven to compete with thyroxine in the attachment to transthyretin [360], which is the main transport protein for thyroid hormones. Therefore, isoflavones might change the concentration of free thyroid hormones, resulting in impaired tissue availability and metabolism, followed by a disturbance in the feedback regulation of hormonal networks, including the pituitary–thyroid–periphery axis during development and in adult organisms. Research conducted by Ariyani et al. [361] has shown a novel mode of action of soybean isoflavones on thyroid hormone receptors (TR) function. They proved that genistein and daidzein augmented T3-liganded TR-mediated transcription in a concentration-dependent manner and augmented the recruitment of steroid receptor coactivator-1 and nuclear corepressor to liganded or unliganded TRs. These findings indicate that the augmentation of the TR-mediated transcription by genistein and daidzein is due to their direct binding to the TR-ligand-binding domain to induce the recruitment of steroid receptor coactivators.

In another study, the correlation was analysed between feeding children with soy-containing formulas and the development of autoimmune thyroid disease [362]. The authors reported that the frequency of feedings with soy-based milk formulas in early life was significantly higher in children with autoimmune thyroid disease as compared to their siblings (prevalence 31% vs. 12%, respectively) as well as to healthy nonrelated control children (prevalence 13%). Therefore, the association of soy formula feedings in infancy with autoimmune thyroid disease was proved.

In conclusion, the available results of various studies are ambiguous and often even contradictory. Further research is needed to clearly show whether the use of isoflavones is safe, or in which subpopulations these compounds will have side effects or could pose a threat.

## 9. Prooxidant Activity of Polyphenols and the Consequences

Although polyphenols are strong antioxidants, in certain circumstances (high level of metals, alkali pH, and O_2_ presence) they can also act as prooxidants. Their prooxidant activity is catalysed mainly by transition metals (copper and iron) and results from the generation of a redox complex with a transition metal ion or a phenoxyl radical [363]. Phenoxyl radicals can then react with oxygen and various reactive oxygen species (ROS) (such as O_2_^−•^ and H_2_O_2_) are formed. These molecules are highly reactive and induce DNA damage or lipid peroxidation, or initiate the oxidation of other important molecules. 

It has been proven that some polyphenols, especially those with small molecules such as dihydroxycinnamic acids, are easily oxidized and cause DNA incision or lipid peroxidation as a result of the activity of radicals produced in the presence of Cu and oxygen [364,365]. Zeng et al. [366] indicated that the compounds bearing a dihydroxyl group in the *ortho*- conformation (e.g., caffeic acid and chlorogenic acid) or bearing a 4-hydroxy-3-methoxyl group (e.g., sinapic acid and ferulic acid) induced significantly higher DNA damage than the ones bearing no such functionalities, and that ROS and Cu ions were involved in the damage. In addition to copper, the metals and metalloids that can catalyse such reactions also include Al, Zn, Ca, Cr, Mn, Co, Ni, Mg, As, and Cd [363].

Taking into account that copper is a rate-limiting nutrient for the growth and proliferation of cancer cells, they are often characterized by higher level of intercellular copper than in normal cells [367]. Therefore, the prooxidant activity of some polyphenols in the presence of copper can be used against cancer cells because they would exert a pro-apoptotic effect in various types of tumour cells, avoiding normal cells [368,369].

There have been many studies demonstrating the prooxidant activity and the associated destructive potential of various polyphenolic compounds, including resveratrol [370], kaempferol [371], curcumin [372,373], caffeic acid [374,375], *p*-coumaric acid [376], ferulic acid [377], gallic acid [378], salicylic, syringic, vanillic, *p*-, *m*- and *o*-coumaric acids, *p*- and *m*-hydroxybenzoic acids [379], catechin [380], epicatechin and epigallocatechin [381], ellagic acid, phloroglucinol, pelargonidin, pelargonidin-3-*O*-rutinoside [382], quercetin [383,384], and metal complexes of quercetin such as quercetin copper(II) complex [385], quercetin nickel(II) complex [386], quercetin zinc(II) complex [387], and quercetin manganese(II) complexes [388]. Moreover, it is not only pure polyphenols which can act as prooxidants. A prooxidant activity was observed for polyphenolic extracts prepared from Syrah and Chardonnay grape pomaces [389] or wines [390]. The extracts made from red raspberry (*Rubus idaeus* L.) stem could exert prooxidant activity when adequate conditions of pH and temperature are provided [391]. In the study by Tsukada et al. [392], an aqueous extract of grape pomace obtained from winemaking upon photo-irradiation generated hydroxyl radicals and, thus, exerted prooxidant activity.

While the prooxidative properties of polyphenols are intended and are a means of destroying cancer cells, polyphenols can be considered beneficial molecules. However, if polyphenols begin to induce the oxidation of normal cell components, including DNA damage, which may eventually lead to mutagenesis, the effects may be undesirable. Therefore, further research should be conducted to explain under what conditions an antioxidant changes its face and begins to be a prooxidant.

## 10. Mutagenic, Cancerogenic and Genotoxic Effects

In some cases, polyphenols can have a negative impact on cells, stimulating mutagenesis or exerting cancerogenic and genotoxic effects. There are several studies which show that some flavonoids can mediate DNA cleavage not only by prooxidant activity, but also due to their interaction with topoisomerase IIα and IIβ. The results showed that genistein enhanced DNA cleavage mediated by human topoisomerase IIα and Iiβ, and the scission could be reversed when EDTA was added to reaction mixtures before cleavage complexes were trapped by SDS [393]. In other studies, (−)-epigallocatechin gallate (EGCG) [394] was proven to be a redox-dependent topoisomerase II poison that acts by covalently adducting to the enzyme. Similar studies have revealed that various polyphenols can inhibit topoisomerase due to different mechanisms. EGCG and (−)-epigallocatechin (EGC) were redox-dependent topoisomerase II poisons, kaempferol and quercetin were traditional poisons, myricetin utilized both mechanisms, and (−)-epicatechin gallate(ECG) and (−)-epicatechin/EC/had no significant activity [395]. According to the authors, the C4′–OH was critical for the polyphenols to act as a traditional poison, while the addition of –OH groups at C3′ and C5′ increased the redox activity of the B ring and allowed the compound to act as a redox-dependent poison. This means that other polyphenols with similar structure can also be potential topoisomerase poisons.

The abovementioned studies were conducted with human cells, and flavonoids were proven to be poisons of human topoisomerase IIα and IIβ. However, there are some data suggesting that flavonoids, such as coumarins, quercetin, EGCG, ECG, EGC, and apigenin, can also act as inhibitors of the bacterial enzymes DNA gyrase and topoisomerase IV [396,397,398,399,400]. Both belong to the type II bacterial topoisomerases; the first is involved in supporting nascent chain elongation during replication of the bacterial chromosome, whereas the second separates the topologically linked daughter chromosomes during the terminal stage of DNA replication [401]. This inhibitory impact on bacterial topoisomerases may result in the dysbiosis of intestinal microbiota, with its all consequences. 

Some polyphenolic compounds can exert mutagenic activity. In the study by Spada et al. [402], 23 samples of frozen fruit were tested for their mutagenic and antimutagenic activity. The obtained results showed that acai, cashew apple, kiwi fruit, and strawberry pulps exhibited mutagenic activity in all loci (*Lys*-revertant, *His*-revertant, and *Hom*-revertant) that were assayed when investigated in a haploid XV 185-14C strain of *Saccharomyces cerevisiae* cells. The impact was observed for high concentrations (5–15% [wt/vol]) and was dose-dependent. Carcinogenic effects of polyphenols can be also a result of their intercalating into DNA or their ability to induce chromosome damage [386,403,404,405]. Some studies suggest that quercetin induces H_2_O_2_-mediated DNA damage, which results in apoptosis or mutations and may be related to the carcinogenic effects of quercetin, whereas luteolin induces apoptosis via DNA cleavage mediated by topoisomerase II [406]. Other data demonstrate that some bioflavonoids can cause chromosomal translocations through either topoisomerase II-dependent (myricetin, genistein, and quercetin) or topoisomerase II-independent (luteolin and kaempferol) mechanisms [405]. Among several dozen naturally occurring flavonoids, the flavones C-glycosides vitexin and orientin showed moderate sister chromatid exchange (SCE)-inducing activity, while other compounds showed only weak activity or were inactive [407]. Procyanidins consisting of three or four flavanol units, and—to a lesser extent—flavone, flavonol, and anthocyanidin aglycones, induced polyploidy. Aglycones, as well as C- and O-glycosides, spiraeoside, and luteolin-7-glucoside, were more or less active in inducing micronuclei in lymphocytes. In turn, no genotoxic effects were caused by flavonol O-glycosides (rutin and hyperoside) as well as monomeric and dimeric flavanols. 

In vitro experiments on calf thymus DNA treated with quercetin for various time periods have shown that the initial interaction of quercetin with DNA may have a stabilizing effect on the DNA secondary structure; however, the prolonged treatment with quercetin led to an extensive disruption of the double helix [408]. When the ability of various flavonoids (morin, apigenin, and naringin) to bind to DNA was investigated, both intercalation and external binding to the DNA duplex were observed [409]. Structural analysis demonstrated that quercetin, kaempferol, and delphinidin can bind weakly to adenine, guanine (major groove), and thymine (minor groove) bases, as well as to the backbone phosphate group; the stability of adduct formation was in the order of quercetin > kaempferol > delphinidin [410]. In further studies, quercetin, kaempferol, and delphinidin were proven to intercalate tRNA duplex, whereas flavonoid-DNA adducts showed both intercalation and external bindings [411]. Low flavonoid concentration induced the stabilization of the double helix, whereas high content caused helix opening. 

In other studies, myricetin [412] and kaempferol [413] were proven to induce a significant concentration-dependent nuclear DNA degradation concurrent with lipid peroxidation, both enhanced by the presence of iron (III) or copper (II). As lipid peroxidation induced by myricetin could be inhibited by SOD in the presence of copper (II), whereas it was enhanced by catalase in the presence of iron (III), similarly for kaempferol, the results suggest prooxidant properties of the studied polyphenols and their dual role in mutagenesis and carcinogenesis.

The UV spectroscopic and fluorescence quenching proved that quercetin can bind to bovine serum albumin (the tryptophan residues are involved) and that the complex may lead to fragmentation of the protein when Cu(II) is present [414].

In vitro studies demonstrated that the exposure of primary human CD34+ hematopoietic cells to biologically relevant concentrations of flavonoids (quercetin, genistein, and kaempferol) caused a dose-dependent double-strand break (DSB) [415]. An incorrect repair of these DSBs resulted in chromosomal translocations and correlated with infant leukaemias. These results were then confirmed in in vivo studies on an animal model, in which heterozygous Atm-ΔSRI mice (DNA repair-deficient mice) obtained a flavonoid-poor (normal) chow or the same chow supplemented with genistein (270 mg/kg) or quercetin (302 mg/kg) throughout pregnancy [416]. The results revealed that prenatal exposure to both quercetin and genistein supplements was associated with higher frequencies of Mll rearrangements and a slight increase in the incidence of malignancies, especially in the presence of compromised DNA repair. One of the most common loci involved in chromosomal translocations is the break point cluster region of the mixed-lineage leukaemia (MLL) gene. This gene is involved in normal haematopoiesis and chromosomal translocations involving the MLL gene are often in various leukaemias. 

Lu et al. [417] have demonstrated that EGCG can induce death and DNA damage in human lung and skin normal cells. The results of their study showed direct evidence of reductive DNA damage in the cells; EGCG at concentrations below 100 µM slightly increased the lung cancer cell viability. Moreover, EGCG induced DNA double-strand breaks and apoptosis in normal cells, and enhanced the mutation frequency. The mutagenic activity of phenolic-rich extracts made from *Ficus adhatodifolia* and *F. obtusiuscula* (plants used in traditional medicine mainly for the treatment of worms) was demonstrated by the *Salmonella typhimurium* reverse mutation test (four strains) [418]. Mutagenic activity of the TA97 strain without metabolic activation was observed for both tested extracts; in the case of the TA102 strain, both extracts were mutagenic with metabolic activation, while the extract of *F. adhatodifolia* was shown to be mutagenic to the TA102 strain without metabolic activation.

The mutagenic properties of the ethanolic extract (EE) from flowers of Combretum leprosum (a traditionally used Brazilian medicinal plant) were proven in experiments performed using the XV185-14c haploid strain of *S. cerevisiae*, while its toxicity was proven in Chinese hamster lung fibroblast (V79) cells [419].

Destructive effects of polyphenols on cells can also be the result of their interactions with cell membranes. Ollila et al. [420] reported that acacetin, rhamnetin, apigenin, and morin caused destabilization of the membrane structure (by the disorientation of the membrane lipids), which resulted in induced leakage from the model vesicle. Similarly, naringenin and hesperidin significantly increased the fluidity of model lipid membranes composed of dimyristoylphosphatidylcholine [421]. Both of the flavonoids induced alterations in the arrangement of polar heads of lipids; hesperidin caused weakly disorder in hydrophobic region, while naringin has an ordering effect in this region. Naringin was also more effective than hesperidin in terms of inducing changes in the membrane organization.

Considering that polyphenols can disrupt the membrane functioning, there is a high risk of their toxic effect on cells. The prooxidative activity of some polyphenols combined with their ability to affect membranes may cause the mitochondrial toxicity due to collapsing the mitochondrial membrane potential and directing the cell into apoptotic pathway [275]. The impaired function of the mitochondria is involved in various human disorders, such as neurodegenerative diseases, diabetes, obesity, and cancer, while disorders in the control of apoptosis may result in carcinogenesis. On the other hand, quercetin, which can interact with DNA, is able to arrest cell cycles and, hence, cause tumour regression by activating the mitochondrial pathway of apoptosis [422]. 

Although the results of in vitro studies or animal models are not always confirmed by in vivo studies on humans, people or companies marketing polyphenol-containing supplements and foods fortified with these compounds should be aware of the dangers associated with polyphenols and their potential negative impact on human health. 

## 11. Conclusions

Analysing the examples cited in this work, it can be seen that many studies have been conducted in vitro, and the doses of polyphenols that exerted a specific effect are often very high and difficult to achieve with the consumption of “normal” food. Unfortunately, in the minds of consumers, there are often only abbreviated versions of the results, such as “improves the condition of the circulatory system,” “reduces cholesterol,” “increases immunity,” “reduces the risk of cancer,” “slimming effect,” etc. Wanting to achieve the intended effect as soon as possible, many reach for a simpler way, i.e., they choose supplements. Thus, the problem of possible negative impacts of polyphenols becomes real, because the consumption of supplements is not controlled in any way, and polyphenol supplements contain huge doses of pure polyphenols. What is more, they are easily available and very cheap. Why are they so dangerous? A few examples to better illustrate the phenomenon are: there are many supplements that contain 2000 mg of pure resveratrol in one pill or capsule; moreover, the recommended serving dose is sometimes more than one piece. Among food, the richest in resveratrol is red wine prepared from muscadine grapes, containing about 30 mg of resveratrol per litre. This means that, to achieve 2 g of resveratrol with wine, one would be required to drink almost 100 L. This is, of course, impossible; however, in the form of supplements, it can be done with ease! Just swallow one capsule. The same situation exists for quercetin. According to the Phenol explorer base (http://phenol-explorer.eu/, accessed on 2 March 2023), the highest concentration in food sources was found in dried oregano, up to 42 mg per 100 g fresh weight. If someone wants to provide their body with 1 g of quercetin, they can either swallow one supplement capsule or consume 2.4 kg of this seasoning. Commercially available supplements (EGCG Green Tea Standardized Extract) contain 1800 mg of green tea extract, in which 45% of the extract constitutes pure EGCG. Infusions of green tea made at home contain about 270 mg EGCG per litre. Again, one can take one capsule, providing the body with about 800 mg of EGCG, or drink approximately 3 L of tea infusion.

It should be clearly emphasized that, in general, polyphenols are safe for healthy individuals when consumed as a part of a balanced and varied diet. They are valuable natural compounds which help to prevent and treat many diseases or defend against various kinds of stress. Therefore, polyphenols should be introduced into our diet, especially as components of natural, low- or non-processed food such as fruit, vegetables, nuts, herbs, spices, tea, or juices. However, the risk of possible negative side effects increases when polyphenols are consumed in large amounts without medical supervision in the form of dietary supplements or as various plant extracts. Such products usually contain huge doses of polyphenols which are often in the form of chemically purified aglycones (instead of glycosides which are present in “natural” food) that are impossible to achieve by normal food consumption. The negative effects of polyphenols are also more likely to occur when a polyphenol-rich diet or supplements are taken by people being treated for various chronic illnesses who are required to take prescribed medications each day. Therefore, the authors of the review claim that it is very important to raise public awareness about the possible side effects of flavonoid supplementation, especially in the case of various vulnerable subpopulations.

Further research is needed to explain in detail which cell types and under what kind of conditions polyphenols can exert prooxidant, mutagenic, or genotoxic effects, and thus promote the development of cancer. It should also be clarified whether the polyphenols may have a beneficial effect due to their damaging impact on the cancer cells only. We claim that, in the near future, all polyphenol supplements, as well as some types of novel food or enriched food, should be provided with detailed characteristics of their ingredients, describing both the positive, desirable influences of their intake on the human body and possible negative impacts, with vulnerable groups being enumerated.

## Figures and Tables

**Figure 1 molecules-28-02536-f001:**
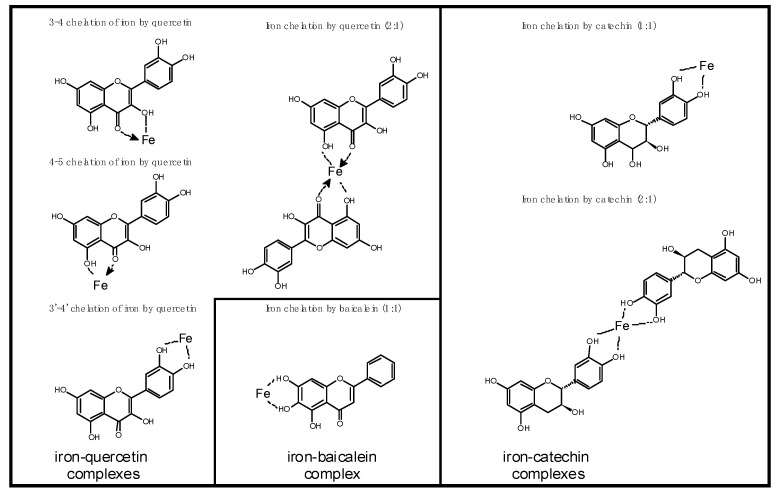
The proposed structures of iron complexes with various flavonoids based on [158,160,161].

**Figure 2 molecules-28-02536-f002:**
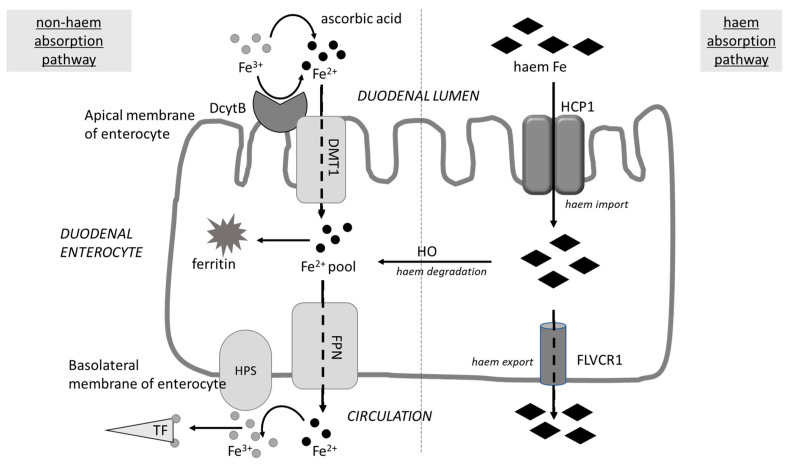
Possible pathways of iron absorption through the duodenal enterocyte based on [161,168,169]. Abbreviations: Fe^3+^—ferric iron, Fe^2+^—ferrous iron, DcytB—duodenal cytochrome B; DMT1—divalent metal transporter 1; FPN—ferroportin; HPS—hephaestin; HCP1—haem carrier protein 1; FLVCR1—feline leukaemia virus subgroup C receptor 1; HO—haem oxidase; TF—transferrin.

**Table 1 molecules-28-02536-t001:** Examples of digestive enzymes inhibited by polyphenols (in vitro studies).

Enzyme	Polyphenol	Method of Evaluation and Results	References
α-amylase	hesperetin (HES), luteolin (LUT), quercetin (QUE), catechin (CAT) and rutin (RUT)	UV–Vis spectroscopy, fluorescence and molecular docking/α-amylase presented a higher affinity for LUT and LUT was better inhibitor than positive control (acarbose), no inhibition was observed with CAT and RUT; docking analysis showed that flavonoids bound near to enzyme active site	[209]
	young apple polyphenols (YAP) and nine types of phenolic compounds	fluorescence quenching/tannic acid, chlorogenic acid, and caffeic acid in YAP showed high inhibition against amylase with the IC50 values of 0.30, 1.96, and 3.69 mg/mL, respectively; the order of the apparent static quenching constants was: tannic acid > chlorogenic acid > caffeic acid > epicatechin	[210]
	various kinds of tea, catechins and theaflavins	green, oolong and black tea extracts, epigallocatechin gallate, theaflavin-3, 3′-digallate, and tannic acid were competitive inhibitors of PPA, whereas epicatechin gallate, theaflavin-3′-gallate and theaflavin were mixed-type inhibitors with both competitive and uncompetitive inhibitory characteristics; only catechins with a galloyl substituent at the 3-position showed measurable inhibition; 3 and 3′ Galloyl substitution increase the inhibitory activity of theaflavins, and increased the association of catechins and theaflavins with amylase	[211]
	sorghum procyanidins (SPC) tetramer	fluorescence, UV-vis absorption, and circular dichroism/SPC-tetramer was bound with human salivary α-amylase at the ratio of 1:1, the conformation of enzyme was altered	[212]
	tea polyphenols	depletion assays, fluorescence spectroscopy, and initial rate kinetics/tea polyphenols inhibited the activity of enzyme and increased the binding rate of porcine pancreatic α-amylase to starch	[213]
	4 caffeic and tartaric acid derivates	inhibition assay, kinetics, fluorescence quenching, and molecular docking/caffeic acid had a low inhibitory activity; however, caffeoyl substitution at 2,3-OH of tartaric acid gradually increased its competitive inhibition character/caftaric acid (one caffeoyl-substituted) and chicoric acid (two caffeoyl-substituted) were suggested as mixed-type and competitive inhibitors, tartaric acid was a typical uncompetitive inhibitor of α-amylase; Fluorescence quenching was only observed for compounds with caffeoyl(s), and the effect increased with the moiety number increasing → caffeoyl moiety entered into α-amylase active pocket.	[214]
	ellagitannins	ellagitannins inhibit α-amylase activity	[215]
	chlorogenic acid (CHA)	kinetic analysis, circular dichroism, fluorescence quenching, and molecular docking/CHA showed a mixed-type inhibitory action on amylase, with the IC50 value of 0.498 ± 0.013 mg/mL; CHA altered the secondary structure of PPA, by interacting with the amino acid residues around or distant from the catalytic site of PPA, mainly through hydrogen bonds, and this interaction was associated with the reduced enzyme’s activity	[216]
	apigenin, scutellarein, hispidulin and nepetin	multispectral methods, fluorescence quenching analysis, and molecular docking/nepetin, a competitive inhibitor, exhibited the best inhibitory effect than other tested flavonoids, suggesting that adjacent dihydroxyl group on the B-ring played an important role in inhibiting the activity of α-amylase.	[217]
α-glucosidase and α-amylase	gallocatechin gallate/GCG/	docking analysis/GCG inhibited α-amylase and α-glucosidase by mixed and non-competitive type. GCG interacted with some amino acid residues located in active site pocket of α-amylase, while it binds to a site close to the active pocket of α-glucosidase GCG form the complexes with enzymes which induced conformational changes	[218]
	epicatechin gallate (ECG)	molecular simulation/ECG inhibited α-amylase/α-glucosidase in a mixed–type manner, it interacted with some residues in the active pocket of enzymes and induced its conformational changes	[219]
	quercetin (1), kaempferol (2), guaijaverin (3), avicularin (4), myricetin (5), hyperin (6) and apigenin (7) isolated from guava leaves	compounds 1, 2, and 5 showed high inhibitory activities, with IC50 values of 3.5 mM, 5.2 mM and 3.0 mM against sucrase, with IC50 values of 4.8 mM, 5.6 mM and 4.1 mM against maltase and with IC50 values of 4.8 mM, 5.3 mM and 4.3 mM against α-amylase, respectively; the hydroxyl group at the 3-position on the A-ring and a number of hydroxyl groups attached to the C-ring played important roles in the inhibition activity	[220]
	extracts of raw and heat-processed (roasted or treated in hot water) African pear (*Dacryodes edulis*)	the extracts inhibited α-amylase activity in a dose-dependent manner; the roasted extract (EC50 = 178.80 μg/mL) had a significantly higher (*p* < 0.05) inhibitory effect on α-amylase activity than the boiled sample (EC50 = 230.45 μg/mL) and the raw sample extract (EC50 = 266.10 μg/mL). The roasted sample (EC50 = 170.94 μg/mL) also had the highest inhibitory effect on the α-glucosidase activity, while the extracts from the raw pear had the least (EC50 = 178.80 μg/mL)	[221]
	herbal extracts containing rosmarinic acid (RA) and purified RA	amylase inhibition correlated with increased concentration of RA; RA-containing oregano extracts yielded higher than expected amylase inhibition than similar amount of purified RA, suggesting that other phenolic compounds or phenolic synergies may contribute to additional amylase inhibitory activity.	[222]
	tea polyphenols (TP) and different types of teas (green, black and oolong tea) processed from the same fresh leaves	all three types of teas significantly enhanced α-amylase activity for a wide range of concentrations (0.34–27.14 mg/mL), and green tea showed the highest activation effect, while high TP concentration slightly inhibited it by non-competitive fashion	[223]
	extracted and enriched flavonoids from *Rubus corchorifolius* (12 isolated flavonoids, 6 of the obtained for the first time)	molecular modelling; flavonoid/compound 4 was the strongest inhibitor of α-glucosidase and α-amylase, to improve postprandial hyperglycaemia	[224]
	ferulic acid (FA)	enzyme kinetic analysis, circular dichroism (CD), Fourier-transform infrared (FT-IR) spectroscopy, fluorescence quenching, and molecular docking; FA inhibited α-amylase/α-glucosidase by mixed/non-competitive mechanisms; secondary structure of enzymes was changed by binding FA and non-covalent bonding was the main force	[225]
α-glucosidases: maltase and sucrase	5-caffeoylquinic acid, EGCG, polyphenol-rich green tea extract/GTE/	GTE efficiently inhibited both human and rat sucrase and maltase activity; 5-caffeoylquinic acid did not significantly inhibit maltase and was only a very weak inhibitor of sucrase.	[226]
	epicatechin-(4β,8)-epicatechingallate (B2-3′-O-gallate), epicatechin gallate (ECG), epicatechin (EC)	inhibition kinetic/IC50 values were as follows: B2-3′-O-gallate (1.73 ± 1.37 µM and 6.91 ± 3.41 µM), ECG (3.64 ± 2.99 µM and 18.27 ± 3.99 µM), and EC (6.25 ± 1.84 µM and 18.91 ± 3.66 µM,) for maltase and sucrase, respectively.	[227]
α-amylase, lactase, maltase, sucrase	flavonols, theaflavins, gallate esters, 5-caffeoylqunic acid and proanthocyanidins	flavonols, theaflavins, gallate esters, 5-caffeoylqunic acid, and proanthocyanidins inhibit α-amylase activity; anthocyanidins and catechin oxidation products, such as theaflavins and theasinsensins, inhibit maltase; sucrase is less strongly inhibited but anthocyanidins seem somewhat effective; lactase is inhibited by green tea catechins.	[228]
lactase (lactase phlorizin hydrolase)	epigallocatechin-3-gallate (EGCG)	EGCG inhibited in vitro hydrolysis of lactose by intestinal lactase and salivary proline-rich proteins (PRPs) shown the protective role against EGCG inhibition of digestive enzymes; inhibition by EGCG of digestive enzymes (α-amylase > chymotrypsin > trypsin > lactase ≫ pepsin) was alleviated ∼2−6-fold by PRPs	[229]
pepsin	caffeic acid (CA)	multi-spectroscopy and MD simulations methods; CA affected both the conformation and the activity of pepsin	[230]
	10 flavonoids	spectroscopic and molecular docking methods/all flavonoids could bind with pepsin to form flavonoid-pepsin complexes and the interaction was spontaneous mainly through electrostatic forces and hydrophobic interactions with one binding site, the interaction resulted in the reduced enzyme activity	[231]
trypsin	quercetin (Q), luteolin (LUT), kaempferol (KMP) and apigenin (APG)	at a concentration of 2.7 mM, inhibition of trypsin (1.6 U/mL) by Q, LUT, KMP and APG was 46.4%, 32.6%, 26.8% and 17.7%, respectively. The interaction of polyphenol-trypsin caused the fluorescence quenching of trypsin and inhibition of radical scavenging activity of flavonoids; the strength of binding depended on the number and position of hydroxyl group of flavonoids and was in decreasing order Q > LUT > KMP > APG	[232]
	various polyphenols	Computer Assisted Drug Design studies/5,7-dihydroxy flavonoid have been found to be a perspective trypsin/trypsin-like-enzyme inhibitor; flavanones and isoflavones are less effective trypsin inhibitors due to a loss of the optimal geometry leading to hydrogen bond interactions; quercetin, myricetin and morin have shown to be the best trypsin inhibitors tested.	[233]
	hesperetin (HES), luteolin (LUT), quercetin (Q), catechin (CAT), and rutin (RUT)	UV-Vis, intrinsic and extrinsic fluorescence spectroscopies, circular dichroism, and molecular docking/flavonoids-trypsin complexes showed static quenching, and QUE and LUT exhibited higher affinity; the hydrophobic interactions between trypsin and flavonoids were predominant; LUT was the best trypsin inhibitor (IC50 = 45.20 ± 1.00 μM)	[234]
pancreatic lipase	methanolic extract of the leaves of *Eremochloa ophiuroides* (centipede grass) containing flavonoids	five the C-glycosidic flavones isolated from the extract showed potent inhibitory effects on pancreatic lipase, with IC50 values ranging from 18.5 ± 2.6 to 50.5 ± 3.9 μM	[235]
α-amylase, α-glucosidase and lipase	total phenolics, total flavonoids and condensed tannin content in crude, semi-purified extracts from 8 types of foods (black tea, green tea, blueberry, blackberry, red cabbage, broccoli, black turtle bean and black soybean) and five fractions from legumes	semi-purified extracts from legumes, tea and berries showed more potency (lower IC50 values) against α-amylase, α-glucosidase than commercial inhibitors; Myricetin showed the highest potency against α-amylase, α-glucosidase and lipase (IC50: 0.38 mg/mL, 0.87 μg/mL and 15 μg/mL, respectively)	[236]
pancreatic lipase (PL), phospholipase A2 (PLA2), and trypsin	tea polyphenols: theaflavin-3,3′-digallate (TFdiG), theaflavin-3′-gallate (TF3′G), theaflavin-3-gallate (TF3G), and theaflavin (TF), catechins, (−)-epigallocatechin-3-gallate (EGCG)	Modelling studies/TfdiG, TF3′G, TF3G, and TF inhibited PL (IC50 = 1.9, 4.2, 3.0, and 32.9 µM, respectively), indicating that the location of the galloyl ester is essential for inhibitory potency; catechins inhibited PL and PLA2; EGCG inhibited trypsin (IC50 = 193 µM) in a non-competitive manner	[237]

**Table 2 molecules-28-02536-t002:** Examples of polyphenol-drug interactions.

Drug (Medical Application)	Polyphenol or Food; Type of Study (Protocol If Known)	Impact on Drug Activity → Conclusions or Recommendations	References
Warfarin (preventing blood clots)	green tea/GT/; in vivo human study (a 44-year-old white man was receiving warfarin for thromboembolic prophylaxis secondary to a St. Jude mechanical valve replacement in the aortic position)	the patient had INR * of 3.20 approximately one month prior to entering the clinic, and an INR of 3.79 on entering the clinic, 22 days later his INR was 1.37, 1 month later the INR was 1.14. It was subsequently discovered that the patient began drinking one-half to one gallon of GT/day about one week prior to the INR of 1.37; discontinuation of the green tea enables the patient’s INR increase to 2.55 → concomitant intake of green tea and warfarin should be under medical supervision	[296]
	resveratrol/RES/; in vivo studies in animal model (rats were orally given (±)warfarin (0.2 mg/kg) without and with RES (100 mg/kg) in a parallel design)	RES significantly increased the AUC_0−t_ of S-warfarin and international normalized ratio. Mechanism is based on the inhibition of BCRP (breast cancer resistance protein)-mediated efflux of R- and S-warfarin. Moreover, RES metabolites activated CYP1A2/3A4, but inhibited CYP2C9 → concomitant intake of RVT increased the systemic exposure of warfarin and enhanced the anticoagulation effect mainly via inhibitions on BCRP and CYP2C9	[297]
	goji berries (*Lycium barbarum* L.) extract; in vivo studies in animal model (4 experimental groups of Wistar rats: distilled water (negative control); fed daily with the extract (0.18 g/kg); treated daily with water and warfarin (0.5 mg/kg—positive control) and those treated concomitantly with the extract and warfarin, for 7 days)	there were no significant differences between the biochemical and haematological profiles, nor even signs of toxicity of the extract when administered alone → concomitant use intake of goji berries extract with warfarin showed a significant increase in prothrombin time, with the potential for bleeding.	[298]
	cranberry; in vivo studies in animal model (rats were orally administered warfarin (0.2 mg/kg) without and with cranberry (5.0 g/kg) at 0.5 h prior to the warfarin, and at 10 h after the warfarin)	cranberry ingested at 0.5 h before warfarin significantly decreased the systemic exposures of S-warfarin and R-warfarin. Conversely, when cranberry was ingested at 10 h after warfarin, the elimination of S-warfarin was significantly inhibited, and the anticoagulation effect of warfarin was significantly enhanced. Probably cranberry activated the breast cancer resistance protein/BCRP/, which mediated the efflux transports of S-warfarin and R-warfarin. The metabolites of cranberry inhibited cytochrome CYP2C9 → the concomitant use of cranberry with warfarin should be avoided.	[299]
	cranberry; in vivo human studies (open-label, three-treatment, randomized crossover clinical trial was undertaken and involved 12 healthy male subjects of known CYP2C9 and VKORC1 genotype)	cranberry significantly increased the area under the INR–time curve by 30% when administered with warfarin compared with treatment with warfarin alone. Cranberry did not alter S- and R-warfarin pharmacokinetics or plasma protein binding. Coadministration of garlic did not significantly alter warfarin pharmacokinetics or pharmacodynamics. Both herbal medicines showed some evidence of VKORC1 (not CYP2C9) genotype-dependent interactions with warfarin, which is worthy of further investigation → Co-administration of warfarin and cranberry requires careful monitoring.	[300]
	cranberry juice; case study	a man in his 70s had a poor appetite for two weeks and ate almost nothing, taking only cranberry juice and his regular drugs (digoxin, phenytoin, and warfarin). Six weeks after starting cranberry juice he had been admitted to hospital with an INR > 50, although before, his control INR was stable. He died due to a gastrointestinal and pericardial haemorrhage → Uncontrolled, concomitant administration of warfarin and cranberry can cause death due to haemorrhage.	[301]
	cranberry juice; case study	a 78-year-old, 86 kg man receiving warfarin at a total weekly dose of 45 mg for atrial fibrillation had INR of 6.45, having reported drinking a half gallon of cranberry/apple juice in the week prior to the elevated INR. After discontinuation of the cranberry juice, maintaining the warfarin dose for 5 days, and resuming the warfarin at a total weekly dose of 40 mg, the INR returned to the therapeutic range of 2 to 3 → combination of warfarin administration and cranberry juice ingestion appeared to be associated with an elevated INR without bleeding in this elderly patient.	[302]
	gouqizi (Goji berry) wine; case study	65-year-old Chinese man taking a prolonged maintenance dose of warfarin who experienced an elevated INR with associated bleeding after drinking Gouqizi wine at large doses → Doctors should advise patients regarding possible interactions between herbs and warfarin when prescribing and should increase the frequency of INR monitoring for those patients concurrently receiving warfarin and medicinal herbs.	[303]
	goji juice; case study	71-year-old Ecuadorean-American woman who was taking warfarin and was hospitalized for a markedly elevated, indeterminate INR (prothrombin time > 120 sec) after consumption of goji juice. She had undergone knee surgery approximately 3 months earlier at which time warfarin therapy was started. She reported no changes in dietary habits or lifestyle other than drinking goji juice for 4 days before hospitalization. On presentation to the emergency department, she described symptoms of epistaxis, bruising, and rectal bleeding → Patients should be educated about avoiding popular herbal drinks or juices, such as goji juice, while they are taking warfarin, while the clinicians should question patients about their use of herbal therapies and document such use in their medical records before prescribing drugs such as warfarin.	[304]
	concentrated Chinese herbal tea made from *Lycium barbarum* L. (goji berry) fruits; case study	a 61-year-old Chinese woman had an elevated INR of 4.1, although before it was stabilized on anticoagulation therapy at level 2–3. There were no changes in her other medications or lifestyle, a review of her dietary habits revealed 4 days of drinking a goji tea (3–4 glasses daily) prior to her clinic visit. After leaving the tea, while maintaining consistency with medications and dietary habits, a follow-up INR seven days later was 2.4, and seven subsequent INR values were in the 2.0–2.5 range → combination of *L. barbarum* L. and warfarin should be avoided.	[305]
Rivaroxaban (prevent blood clots)	naringenin; in vitro (liver microsomes); in vivo animal model (male Sprague–Dawley rats were randomly divided into the experimental (Ex) group and the control (C) group with six rats in each group; Ex rats were pre-treated with naringenin (10 mg/kg/day) for 2 weeks before the administration of rivaroxaban (10 mg/kg) by oral gavage, while the C rats were given rivaroxaban (10 mg/kg) only once)	(i) in vitro data indicated that naringenin could decrease the metabolic clearance rate of rivaroxaban with the IC50 value of 38.89 μM, and exhibited a mixed inhibition to rivaroxaban; (ii) compared to C group the AUC_0–t_ value was increased in Ex rats from 2406.28 ± 519.69 μg/h/L (in controls) to 4005.04 ± 1172.76 μg/h/L, the C_max_ value was increased from 310.23 ± 85.76 μg/L to 508.71 ± 152.48 μg/L, and the V_z/F_ and CL_z/F_ were decreased from 23.03 ± 4.81 L/kg to 16.2 ± 8.42 L/kg, 4.26 ± 0.91 L/h/kg to 2.57 ± 0.73 L/h/kg, respectively → naringenin had an inhibitory effect on the pharmacokinetics of rivaroxaban in rats	[306]
Saquinavir (protease inhibitor used for HIV infection treatment)	garlic supplement; in vivo human study (10 healthy volunteers received 10 doses of saquinavir (Fortovase) at a dosage of 1200 mg, 3 times daily with meals for 4 days on study days 1–4, 22–25, and 36–39, and they received a total of 41 doses of garlic caplets taken 2 times daily on study days 5–25.)	in the presence of garlic, the mean saquinavir area under the curve (AUC) during the 8-h dosing interval decreased by 51%, trough levels at 8 h after dosing decreased by 49%, and the mean maximum concentrations (C_max_) decreased by 54%. After the 10-day washout period, the AUC, trough, and C_max_ values returned to 60–70% of their values at baseline → Patients should use caution when combining garlic supplements with saquinavir when it is used as a sole protease inhibitor	[307]
Metformin (antihyperglycemic agent used for the treatment of type 2 diabetes, particularly in people who are overweight)	green tea (GT) and EGCG; in vitro studies	(i) metformin uptake was inhibited in a concentration-dependent manner in the presence of GT with IC_50_ values of 1.4% (*v*/*v*) and 7.0% (*v*/*v*) for OCT1 and OCT2, respectively; (ii) the inhibitory potency of GT on metformin uptake was stronger for MATE1 compared to MATE2-K; (iii) IC_50_ of green tea was 4.9% (*v*/*v*) for inhibition of MATE1-mediated metformin transport, while the IC_50_ value for MATE2-K-mediated metformin transport could not be calculated; (iv) OCT1-mediated metformin net uptake (i.e., uptake into transporter-transfected cells minus uptake into vector control cells) was significantly reduced by EGCG to 40% of metformin net uptake without EGCG; (v) OATP1B1-mediated BSP and atorvastatin net uptake (i.e., uptake into transporter-transfected cells minus uptake into vector control cells) were reduced by EGCG to 64% (not significant) and 69% (*p* < 0.05), respectively, of net uptake without EGCG; (vi) the GT significantly decreased the basal-to-apical digoxin transport to 2.4%/h for 1% (*v*/*v*) green tea → green tea and its main catechin ECGC inhibit in vitro transport of prototypical substrates of all seven drug transporters investigated (OCT1, OCT2, MATE1, MATE2-K, OATP1B1, OATP1B3, P-gp).	[293]
	silibinin, epigallocatechin (ECGC), quercetin and rutin; in vivo animal study (30 male rats were divided into 5 groups and treated as follow: control group treated with olive oil (0.2 mL/day); the other 4 groups were treated with either silibinin (100 mg/kg), ECGC (25 mg/kg), quercetin (50 mg/kg) or rutin (500 mg/kg), administered orally as oily solutions for 30 days. At day 30, a 300 mg/kg metformin and 50 mg/kg atenolol were administered orally)	all polyphenols produced significant increase (*p* < 0.05) in serum levels of metformin compared with control group, while atenolol levels revealed no significant differences compared with controls, except for silibinin for which significant increase was reported. Silibinin and EGCG long-term use produced significant increase in metformin contents in bran and kidney, and for EGXG also in the liver → Long-term administration of silibinin, EGCG, quercetin or rutin increase oral absorption and tissue distribution of metformin, while atenolol was not affected.	[308]
Digoxin (the oldest medication used to treat various heart conditions, most frequently for atrial fibrillation, atrial flutter, and heart failure)	green tea; in vivo human study (0.5 mg of digoxin was administered orally to 16 healthy volunteers at Day 1, after a 14-day washout period, 630 mg of green tea catechins/GTC/was administered via oral route, followed by 0.5 mg of digoxin 1 h later; from Day 16 through Day 28, 630 mg of GTC was administered alone; At Day 29, 630 mg of GTC and 0.5 mg of digoxin were administered in the same way as Day 15)	compared to digoxin alone, the concomitant administration of digoxin and GTC significantly reduced the systemic exposure of digoxin: geometric mean ratios/GMR/of area under the concentration–time curve from time 0 to the last measurable time/AUC_last_/and C_max_ were 0.69 and 0.72, respectively. The concomitant administration of digoxin and GTC following pretreatment of GTC (Day 29) similarly reduced the AUC_last_ (GMR = 0.67) and C_max_ (GMR = 0.74) → the coadministration of GTC reduces the systemic exposure of digoxin regardless of pretreatment of GTC.	[309]
Midazolam (a benzodiazepine medication used for anaesthesia and procedural sedation, to treat severe agitation and insomnia)	green tea extract/GT/ and grape seed extract/GSE/; in vitro and in vivo studies in animal model (3 groups of rats with single administration of herbal extract, GTE 400 mg/10 mL b.w.; GSE 80 mg/kg b.w, water-control 10 mL/kg b.d. administered orally after overnight fasting; 3 groups of rats with subchronic treatments: GTE, GSE and water as above but daily administered for 6 successive days)	strong inhibition of these CYP2C9, CYP2D6, and CYP3A4 activities in human liver microsomes by GTE and GSE in vitro; in rats, single treatments with these extracts had negligible effects, 1 week of GTE/GSE treatment resulted in significantly increased elimination rate constant (ke) of intravenously administered midazolam/MDZ/, indicating the induction of CYP3A in the liver. In contrast, 1 week of treatment with GTE, but not GSE, caused a significant increase in the C_max_ and AUC_∞_ of orally administered MDZ without change in the elimination half-life, suggesting a reduction in CYP3A activity in the small intestines → subchronic ingestion of GTE or GSE may alter the pharmacokinetics of midazolam, the effects of GTE on CYP3A activity appear opposite between liver and small intestine.	[310]
	grapefruit juice; in vivo human study (8 healthy male subjects have administered midazolam/MDZ/intravenously (5 mg) or orally (15 mg) after pretreatment with water or grapefruit juice)	after intravenous administration no changes in the pharmacokinetics or pharmacodynamics of MDZ. After oral administration of MDZ: pretreatment with grapefruit juice led to a 56% increase in peak plasma concentration (C_max_), a 79% increase in time to reach C_max_ (t_max_), and a 52% increase in the area under the plasma concentration-time curve (AUC) of MDZ, which was associated with an increase in the bioavailability from 24% ± 3% (water) to 35% ± 3% (grapefruit juice, *p* < 0.01); was also associated with a 105% increase in t_max_ and with a 30% increase in the AUC of alpha-hydroxyMDZ → pretreatment with grapefruit juice is associated with increased bioavailability and changes in the pharmacodynamics of midazolam that may be clinically important, particularly in patients with other causes for increased midazolam bioavailability such as advanced age, cirrhosis of the liver, and administration of other inhibitors of cytochrome P450.	[311]
Sildenafil (a medication used to treat erectile dysfunction and pulmonary arterial hypertension) + midazolam	green tea (GT); in vivo human study (each of 10 healthy volunteers received one tablet of sildenafil 50 mg and one tablet of midazolam 7.5 mg concurrently either after drinking 250 mL of water or 250 mL of fresh extract of 2 g of green tea; after 1 week washout period, each volunteer received the other intervention)	(i) coadministration of GT with sildenafil increased the extent but not the rate of sildenafil absorption, which resulted in higher plasma concentrations (AUC_∞_ increased from 484.2 ± 67.27 μg hr/L to 731.5 ± 111.01 μg hr/L and the C_max_ from 318.9 ± 46.8 μg/L to 414.9 μg/L ± 67.0 μg/L; (ii) the elimination rate constant of sildenafil was significantly decreased and the elimination half-life was prolonged by about 36%; (iii) The AUC_∞_ of midazolam increased by 16% and C_max_ by 14%; suggesting a small reduction CYP 3A4 activity. → Patients who are taking green tea may need smaller doses of sildenafil, and those at higher risk of developing sildenafil adverse effects	[312]
Fluvastatin (belongs to statins, HMG-CoA reductase inhibitors, a class of lipid-lowering medications that reduce illness and mortality in people of high risk of cardiovascular disease; the most common cholesterol-lowering drugs)	green tea/GT/, (−)-epigallocatechin gallate/EGCG/; in vitro study (Bactosomes prepared from *Escherichia coli* cells coexpressing recombinant human NADPH-P450 reductase and human CYP2C9) in vivo human studies (11 healthy volunteers ingested a single 20-mg dose of fluvastatin with: (1) 300 mL of brewed green tea; (2) 150 mg of EGCG in 300 mL of water/GTE/; or (3) 300 mL of water as control, after overnight fasting)	in vitro EGCG inhibited fluvastatin degradation with IC50 of 48.04 μM. Brewed green tea used in the clinical study also dose-dependently inhibited the metabolism of fluvastatin in vitro. No significant effects of GTE and brewed green tea were observed in plasma concentrations of fluvastatin. The geometric mean ratios with 90% CI for area under the plasma concentration-time curve (AUC_0−∞_) of fluvastatin were 0.993 (brewed green tea) and 0.977 (GTE) → although in vitro studies indicated that EGCG and brewed green tea produce significant inhibitory effects on CYP2C9 activity, the concomitant administration of green tea and fluvastatin in healthy volunteers did not influence the pharmacokinetics of fluvastatin	[313]
Atorvastatin (statin)	green tea extract; in vivo human study (12 healthy volunteers received a single dose of atorvastatin 40 mg alone (control group), atorvastatin 40 mg plus a capsule containing 300 mg of dry green tea extract/GTE300/, or atorvastatin 40 mg plus a capsule containing 600 mg of dry green tea extract/GTE600/)	compared to control, the GTE300 and GTE600 decreased the peak plasma concentration (C_max_) of atorvastatin by 25% and 24%, respectively (*p* < 0.05), and the area under the plasma concentration-time curve (AUC_0−∞_) of atorvastatin by 24% and 22%, respectively (*p* < 0.05); it also increased the apparent oral clearance of atorvastatin by 31% and 29%, respectively. The T_max_ and the elimination half-life of atorvastatin did not differ among the three phases. The effects of GTE600 on the pharmacokinetic parameters of atorvastatin were not significantly different from GTE300 → Green tea extract decreases the absorption but not the elimination of atorvastatin, possibly by inhibiting OATP, albeit not in a dose-dependent manner. Coadministration of GTE with atorvastatin may necessitate the monitoring of the drug level in blood.	[314]
Simvastatin (statin)	green tea/GT/and soy isoflavones/SIF/; in vivo human study (18 healthy Chinese male volunteers obtained a single dose of 20 mg simvastatin three times: 1. simvastatin only; 2. with green tea extract; 3. with soy isoflavones extract. There was a washout period of at least 4 weeks between phases. The green tea and soy isoflavone extracts were given at a dose containing EGCG 800 mg once daily or soy isoflavones about 80 mg once daily for 14 days before simvastatin dosing)	SIF intake was associated with reduced systemic exposure to simvastatin acid (AUC_0–24 h_ from 16.1 h∙mg/L to 12.1 h∙mg/L, *p* < p0.05), but not the lactone. The interaction between simvastatin and SIF only resulted in a significant reduction of AUC in subjects with the SLCO1B1 521TT genotype and not in those with the 521C variant allele. There was no effect of GTE on simvastatin pharmacokinetics, only the group with the SLCO1B1 521TT genotype showed reduced AUC values for simvastatin acid → repeated administration of soy isoflavones reduced the systemic bioavailability of simvastatin in healthy volunteers, which was dependent on the SLCO1B1 genotype suggesting that SIF-simvastatin interaction is impacted by genotype-related function of this liver uptake transporter.	[315]
Rosuvastatin (statin)	green tea/GT/; in vivo human studies (healthy volunteers aged 20–55 years received a 20-mg rosuvastatin tablet with 150 mL of water by oral route on Day 1. After a 3-day washout period, they received 300 mg of EGCG followed by 20 mg of rosuvastatin 1 h later. From Day 5 through 14, subjects only received 300 mg of EGCG. On Day 15, just like Day 4, they received 300 mg of EGCG followed by 20 mg of rosuvastatin 1 h later)	compared with the administration of rosuvastatin alone, the concomitant use at Day 4 significantly reduced the area under the concentration–time curve from time 0 to the last measurable time (AUC_last_) by 19% (geometric mean ratio 0.81, 90% confidence interval [CI] 0.67–0.97) and the peak plasma concentration (C_max_) by 15% (geometric mean ratio 0.85, 90% CI 0.70–1.04). AUC_last_ or C_max_ of rosuvastatin on Day 15 was not significantly different from that on Day 1 → Coadministration of EGCG reduces the systemic exposure of rosuvastatin by 19%, and pretreatment of EGCG can eliminate that effect of co-administration of EGCG.	[316]
	green tea extract/GTE/and (–)-epigallocatechin-3- gallate/EGCG/; in vitro (Caco-2 cells and OATP1B1-HEK293T cells) in vivo animal studies	in the Caco-2 cell model, the uptake and transport of rosuvastatin in the GTE groups were 1.94-fold (*p* < 0.001) and 2.11-fold (*p* < 0.050) higher, respectively, than those of the control group. However, in the EGCG group, the uptake and transport of rosuvastatin were decreased by 22.62% and 44.19%, respectively (*p* < 0.050). In the OATP1B1-HEK293T cell model, the OATP1B1-mediated rosuvastatin uptake was decreased by GTE to 35.02% of that in the control (*p* < 0.050) and was decreased by EGCG to 45.61% of that in the control (*p* < 0.050). GTE and EGCG increased the AUC_0−∞_ of rosuvastatin (*p* < 0.050) → GTE increased the systemic rosuvastatin exposure in rats, and the mechanism may include an increase in rosuvastatin absorption and a decrease in liver distribution by inhibiting OATP1B1	[317]
	honey flavonoids (galangin, myricetin, pinocembrin, pinobanksin, chrysin and fisetin); in vitro study (cell lines overexpressing the hOATP2B1 or hOATP1A2 transporter)	chrysin, galangin, and pinocembrin inhibited both hOATP2B1 and hOATP1A2 in lower or comparable concentrations as the known flavonoid OATP inhibitor quercetin. Galangin, chrysin and pinocembrin effectively inhibited rosuvastatin uptake by hOATP2B1 with IC50 ∼1–10 μM. The inhibition of the hOATP1A2-mediated transport of rosuvastatin by these flavonoids was weaker. → several natural flavonoids present in honey can affect drug cellular uptake by hOATP2B1 and/or hOATP1A2 at relative low concentrations suggesting the possibility of food-drug interactions.	[318]
	green tea extract/GTE/ and soy isoflavonoids/SIF/; in vivo human study (20 healthy Chinese males were given a single dose of rosuvastatin 10 mg three times: 1. rosuvastatin alone; 2. with GTE; 3. with SIF. The GTE and SIF were given at a dose containing EGCG 800 mg once daily or soy isoflavones 80 mg once daily for 14 days before statin dosing and at the same time as the statin dosing with at least 4-weeks washout period between phases).	GTE intake significantly reduced the systemic exposure to rosuvastatin by about 20% reducing AUC_0–24 h_ from 108.7 h·μg/L to 74.1 h·μg/L and C_max_ from 13.1 μg/L to 7.9 μg/L (*p* < 0.001 for both), without affecting the elimination half-life. SIF had no significant effect on rosuvastatin pharmacokinetics. → repeated administration of GTE significantly reduced the systemic exposure of rosuvastatin in healthy volunteers.	[319]
Nadolol (β-blocker)	green tea extract/GTE/and (–)-epigallocatechin-3-gallate/EGCG/; in vivo animal model (Male Sprague-Dawley rats received GTE (400 mg/kg), EGCG (150 mg/kg) or saline (control) by oral gavage, 30 min before a single intragastric administration of 10 mg/kg nadolol)	pretreatment with GTE resulted in marked reductions in the C_max_ and AUC of nadolol by 85% and 74%, respectively, as compared with control. In addition, EGCG alone significantly reduced C_max_ and AUC of nadolol. Amounts of nadolol excreted into the urine were decreased by pretreatments with GTE and EGCG, while the terminal half-life of nadolol was not different among groups. → the coadministration with GT catechins, particularly EGCG, causes a significant alteration in the pharmacokinetics of nadolol, possibly through the inhibition of its intestinal absorption mediated by uptake transporters.	[320]
	green tea/GT/; in vitro study (human embryonic kidney 293) and in vivo human studies (10 healthy volunteers received a single oral dose of 30 mg nadolol with GT or water after repeated consumption of 700 mL GT/day or water for 14 days)	GT markedly decreased C_max_ and AUC_0–48_ of nadolol by 85.3% and 85.0%, respectively (*p* < 0.01), without altering renal clearance of nadolol. The effects of nadolol on systolic blood pressure were significantly reduced by green tea. [3H]-Nadolol uptake assays in human embryonic kidney 293 cells stably expressing the organic anion–transporting polypeptides OATP1A2 and OATP2B1 revealed that nadolol is a substrate of OATP1A2, but not of OATP2B1 and that GT significantly inhibited OATP1A2-mediated nadolol uptake → These results suggest that green tea reduces plasma concentrations of nadolol possibly in part by inhibition of OATP1A2-mediated uptake of nadolol in the intestine.	[321]
	(−)-epigallocatechin-3-gallate/EGCG/; in vivo animal study (male rats aged 12–13 weeks were divided into 4 groups: control, EGCG (pretreated 14 days with EGCG), nadolol (received single dose of nadolol), and EGCG-nadolol (pretreated 14 day with EGCG and received a single dose of nadolol). EGCG (10 mg/kg body weight/day) was given orally for consecutively 13 days at the same time of the day. The rats were fasted for a night and on Day 14, a single dose of nadolol (10 mg/kg body weight) was given orally 30 min after the last dose of EGCG administration)	systolic blood pressure (SBP) of rats EGCG-nadolol was significantly higher than in those which received nadolol alone. Pre-treatment of EGCG resulted in a marked reduction of C_max_ and AUC by 53% and 51%, respectively, compared to control. → exposure to EGCG lead to reduced nadolol bioavailability and therefore, uncontrolled raised blood pressure and higher risks of cardiovascular events.	[322]
	(−)-epigallocatechin-3-gallate/EGCG/; in vivo human studies (3 healthy volunteers received single doses of 30 mg nadolol orally with water (control), or an aqueous solution of EGCG-concentrated green tea extract/GTE/at low or high dose)	a single coadministration of low- and high-dose GTE significantly reduced the plasma concentrations of nadolol, AUC_0–∞_ of nadolol (0.72 for the low and 0.60 for the high GTE dose). There were no significant differences in T_max_, elimination half-life, and renal clearance between GTE and water phases. No significant changes were observed for blood pressure and pulse rate between phases. EGCG competitively inhibited OATP1A2-mediated uptake of sulphobromophthalein and nadolol with Ki values of 21.6 and 19.4 μM, respectively → due to EGCG even a single coadministration of green tea may significantly affect nadolol pharmacokinetics.	[323]
	green tea/GT/; in vivo human studies (1 healthy volunteers received an oral administration of nadolol with, or 1 h after pre-ingestion of brewed GT, or with water in a volume of 150 mL)	in control group AUC_0–48_ of nadolol was 830.5 h∙ng/mL, concomitant GT ingestion and GT ingestion 1 h before nadolol administration resulted in a significant reduction of AUC_0–48_ to 359.0 and 453.6 h∙ng/mL, respectively. There were no differences in time to maximal plasma concentration and renal clearance of nadolol among groups → single concomitant ingestion of GT substantially decreases plasma concentrations of nadolol. Moreover, the reduction in nadolol bioavailability could persist for at least 1 h after drinking a cup of GT	[324]
Lisinopril (highly hydrophilic long-acting angiotensin-converting enzyme inhibitor, is frequently prescribed for the treatment of hypertension and congestive heart failure)	aqueous solution of EGCG; in vivo human studies (10 healthy subjects ingested 200 mL of an aqueous solution of GTE containing ~300 mg of EGCG, or water (control) when receiving 10 mg of lisinopril after overnight fasting)	lisinopril C_max_, AUC_0–24_, and AUC_0–∞_ in the GTE phase were significantly decreased by 71% (*p* < 0.001), 69% (*p* < 0.001), and 67% (*p* < 0.001), respectively, compared with values in the control phase. The geometric mean ratio (GTE/control) for C_max_ and AUC_0–∞_ of lisinopril were 0.289 and 0.337, respectively. No significant differences were observed in T_max_ and renal clearance of lisinopril → the extent of intestinal absorption of lisinopril was significantly impaired in the presence of GTE, whereas it had no major effect on the absorption rate and renal excretion of lisinopril	[325]
Diltiazem (calcium channel blockers)	morin; in vivo animal study (rats were orally administrated with diltiazem (15 mg/kg) in the presence and absence of morin at various concentrations (1.5, 7.5 and 15 mg/kg)	compared to the control given diltiazem alone, the C_max_ and AUC of diltiazem increased by 30–120% in the rats co-administered with a 1.5 or 7.5 mg/kg of morin, while there was no significant change in T_max_ and terminal plasma half-life of diltiazem in the presence of morin. Therefore, absolute and relative bioavailability values of diltiazem in the rats co-administered with morin were significantly higher (*p* < 0.05) than those from the control group → morin significantly enhanced the oral exposure of diltiazem, suggesting that concurrent use of morin or morin-containing dietary supplement with diltiazem should require close monitoring for potential drug interactions.	[326]
	resveratrol/RES/; in vivo animal study (rats were divided into groups with oral administration of 15 mg/kg of diltiazem dissolved in water (3.0 mL/kg) without (control) or with 0.5, 2.5, and 10 mg/kg of resveratrol (mixed in distilled water; total oral volume of 3.0 mL/kg); an intravenous group had injected 5 mg/kg of diltiazem, total injection volume of 1.5 mL/kg).	RES presence significantly (*p* < 0.05) increased AUC of diltiazem, except for resveratrol 0.5 mg/kg, compared to the control group, therefore the absolute bioavailability of diltiazem in the presence of 2.5 and 10 mg/kg RES was significantly higher (10.2–11.1%) than that of the control (6.9%). The relative bioavailability of diltiazem in the presence 2.5 and 10 mg/kg RES was increased by 1.48- to 1.60-fold, respectively. RES did not change absorption rate constant and T_max_ of diltiazem. → resveratrol significantly increased the bioavailability of diltiazem due to the inhibition of both the cytochrome P450 (CYP) 3A4-mediated metabolism and the efflux pump P-glycoprotein (P-gp) in the intestine and/or liver.	[327]
Amlodipine (calcium channel blockers)	(−)-epigallocatechin-3-gallate/EGCG/; in vivo animal study (rats had orally administered amlodipine (1 mg/kg) with or without EGCG pretreatment at the dose of 30 mg/kg/day for 10 days)	rats pretreated with EGCG had the C_max_ of amlodipine increased from 16.32 ± 2.57 to 21.44 ± 3.56 ng/mL (*p* < 0.05), the T_max_ decreased from 5.98 ± 1.25 to 4.01 ± 1.02 h (*p* < 0.05), the AUC_0−t_ increased from 258.12 ± 76.25 to 383.34 ± 86.95 μg∙h/L (*p* < 0.05), and the metabolic half-life was prolonged from 31.3 ± 5.6 to 52.6 ± 7.9 min (*p* < 0.05), suggesting that EGCG affected the pharmacokinetic behaviour of amlodipine → the drug-drug interaction between EGCG and amlodipine might occur, due to the metabolism inhibition of amlodipine by EGCG when they were co-administered.	[328]
Verapamil (calcium channel blockers)	(−)-epigallocatechin-3-gallate/EGCG/; in vivo animal study (9 mg/kg verapamil was administered orally to Sprague-Dawley rats 30 min after the oral administration of 2 and 10 mg/kg of oral EGCG)	compared with the controls, the AUC value of verapamil were greater in the presence of EGCG (74.3% and 111% increase for 2 and 10 mg/kg EGCG, respectively) → probably the inhibition of P-glycoprotein was the mechanism.	[329]
Felodipine (calcium channel blockers)	grapefruit juice; in vivo human study (10 healthy men were given 8 oz of grapefruit juice 3× a day for 6 days. Before and after receiving grapefruit juice, small bowel and colon mucosal biopsies were obtained endoscopically, oral felodipine kinetics were determined, and liver CYP3A4 activity was measured)	grapefruit juice did not affect liver CYP3A4 activity, colon levels of CYP3A5, or small bowel concentrations of P-glycoprotein, villin, CYP1A1, and CYP2D6. In contrast, the concentration of CYP3A4 in small bowel epithelia (enterocytes) fell 62% (*p* < 0.001) with no corresponding change in CYP3A4 mRNA levels. Enterocyte concentrations of CYP3A4 measured before grapefruit juice consumption correlated with the increase in C_max_ when felodipine was taken with grapefruit juice → mechanism of the impact of grapefruit juice on oral felodipine kinetics is its selective downregulation of CYP3A4 in the small intestine.	[330]

* INR—an international normalized ratio.

## Data Availability

Not applicable.

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
