# Peer review of "Possible Side Effects of Polyphenols and Their Interactions with Medicines"

_molecules, 2023, doi:10.3390/molecules28062536_

Round 1

Reviewer 1 Report

Title of the manuscript: “Negative impact and side effects of polyphenols”  

Authors: Aleksandra Duda-Chodak  and Tomasz Tarko 

 Manuscript ID: molecules-2252496

Type of manuscript: Review

The manuscript deals with the state of the art of knowledge on the negative impact of polyphenols on human health, considering the possible side effects related to polyphenol intake.

The review article is very well organized: it contains a new point of view on polyphenols, polyphenol-containing supplements and foods fortified. The authors showed that there is a need to raise public awareness about the possible side effects of the bio-phenol supplementation, especially in the case of vulnerable subpopulations.

1. The manuscript is scientific. The article is very informative and objective.

2. The main object of the paper was illustrated very clearly throughout the text.

The review begins by dealing with the physiological role of polyphenols and their potential use in disease prevention, followed by a detailed description in individual chapters of the harmful effects of polyphenols exerted in particular situations (their capacity to block iron absorption, inhibition of enzymes digestive systems, inhibition of intestinal microflora, interactions with drugs and their impact on hormonal balance). Finally, the authors report the description of the prooxidative activity of the polyphenols and the mutagenic, carcinogenic and genotoxic effects.

The paper addresses the main question clearly and as I explained before it is very interesting and relevant.

As illustrated by the authors in lines 11-14 and 894-903 the topic is original.

Finally, the general consideration of authors about the balance between the positive and side effects of polyphenols intake for people and the future research plan (lines 894-907) should be added as a separate chapter at the end of the review. 

Overall, I appreciate the wide number of references and the organization of the tables which summarize most of the collected data but, taking into account the comments in the attached file, I suggest a minor revision for English style before publication in Molecules. (see also the comments in the pdf-file)

3. The following articles should be cited in the introduction in lines 41-45:

https://onlinelibrary.wiley.com/doi/full/10.1002/jsfa.12229

https://www.mdpi.com/2076-3921/12/1/185

https://link.springer.com/article/10.1007/s40279-018-0998-x

https://analyticalsciencejournals.onlinelibrary.wiley.com/doi/epdf/10.1002/mas.21744?src=getftr

Author Response

Our answers are in the attached file

Reviewer 2 Report

In my opinion, the submitted manuscript „Negative impact and side effects of polyphenols” meets aims and scope of „Molecules” Journal, Special Issue „Biological Activity of Phenolics and Polyphenols in Nature Products” but may be accepted only after major revision.

1.       In my opinion, a more appropriate title for this work would be „Side effects and interactions with medicines of (dietary?) pholyphenols”. Side effects are negative by definition. The same applies to the statement „unpleasant side effects” (line 152) – remove the word „unpleasant”. What is more, the interactions are not side effects (it is a different part of the medicine leaflet).

2.       The authors should lay out the accents differently, the manucript in this shape may be confusing and misleading for readers not familiar with the topic of polyphenols (e.g. Inhibition of intestinal microbiota: line 413 „the negative impact”; while line 500 „positive influence” and line 522 „dysbiosis” Does eating polyphenols causes IBS?). Authors often cite the results of in vitro studies or reports of individual  „unusual” human cases (e.g.line 662 or 631). The results of clinical trials in humans are rather missing. Population studies show that the consumption of plant foods is beneficial for health. There are many publications about health benefits of drinking green tea, or explaining the relationship between resveratrol and the „French paradox”. The conclusions drawn by the authors from the results of in vitro or animal studies should not be so general and far-reaching. It would also be necessary to explain the differences in the dose taken in the case of eating foods containing polyphenols and the doses that are administered during research on cells or animals. In addition, some of the described biological activities of polyphenols are difficult to unambiguously classify as harmful. For example, is inhibiting digestive enzymes always negative? Maybe it causes longer digestion of plant food, which can also lower the glycemic index of the eaten meal? Some interpretation of the presented results is needed. The reader may draw different conclusions.

3.       I disagree with line 41: „mainly by plant food”. We can not forget about drinks as a source of these compounds (green and black tea, red wine). It should also be emphasized that the sources of polyphenols in the human diet are very closely related to the place of residence in the world and the possible consumption of traditional dishes, e.g. Asian Cheonggukjang, Chinese green tea, French and Italian red wine, but generally, the most important dietary sources of flavonoids are green tea, red wine, fruits and vegetables.

4.       Moreover, in the introduction, when listing the effect of polyphenols on humans, it should be mentioned that they are activators of sirtuins (e.g. SIRT-1), which may be related to their beneficial for health effect on humans.

5.       According to the PRISMA guidelines (http://prisma-statement.org/PRISMAStatement/Checklist.aspx) authors should: Specify the inclusion and exclusion criteria for the review and how studies were grouped for the syntheses. Specify all databases, registers, websites, organisations, reference lists and other sources searched or consulted to identify studies. Present the full search strategies for all databases, registers and websites, including any filters and limits used. Specify the methods used to decide whether a study met the inclusion criteria of the review, including how many reviewers screened each record and each report retrieved, whether they worked independently, and if applicable, details of automation tools used in the process…  The manuscript does not describe how the literature for this review was collected.

6.       I believe that the title of second subsection should be changed, because polyphenols are xenobiotics for the human body, which by definition cannot have physiological functions. They perform such in plants, where they occur naturally. If the authors meant the physiological functions of polyphenols in plants, it should be written in such a way that there is no doubt (in this case, however, more functions of these compounds than just antioxidant ones should be described).

7.       Neither quercetin and baicalein nor catechin compound contain Fe (figure 1.), the caption in the figure should be more accurate, e.g. iron-quercetin complex (not quercetin, baicalein, catechin).

8.       Publication 205 („Screening bifunctional flavonoids of anti-cholinesterase and anti-glucosidase by in vitro and in silico studies: Quercetin, kaempferol and myricetin”) cited in table 1 does not meets the criteria of this subsection. It is some kind of a mistake. The inhibition of acetylcholinesterase (in the brain) by polyphenols is an example of a mechanism of action identical to drugs used in treat Alzheimer's disease (e.g. donepezile). Authors conclude, that diet rich in polyphenols protect against dementia and type 2 diabetes. This is an example of health-beneficial enzyme inhibition.

9.       There is no abbreviation list in the publication – please complete it.

10.   Line 273, the Latin name of a plant should be in italics.  The same applies to the Latin terms in vitro and in vivo throughout the manuscript.

11.    Line 278: add the number of figure.

12.   Line 399: no capital letter at the beginning of the sentence

13.   Line 540: drugs are xenobiotics.

14.   Line 894: this should be the conclusion section.

Author Response

Our answers are in attached file.

Reviewer 3 Report

This paper presented a review of the impact of polyphenols on human health. In the beginning, it described the physiological role of polyphenols and their beneficial role in human health. Then, it tried to report the possible side effects of polyphenol intake in “particular situation”. 

The last sentence in Abstract says “According to the authors there is a need to raise public awareness about the possible side effects of flavonoid supplementation, especially in the case of vulnerable subpopulations.” 

Based on this statement and the following observations, it seems to me that the structure of the manuscript was wrongly built, the title is deceiving and the whole paper, as it stands, can be damaging to healthy eating including polyphenols from natural sources and their beneficial effects on human health.

4. Blocking iron uptake

The title of this section should be changed, as it talks and ends with the beneficial effects of polyphenols on iron

6. Inhibition of intestinal microbiota and consequences

Again, the title of this section is deceiving, not mentioning that most of the studies were in vitro. Yes, the gut microbiota can be inhibited in a dose-dependent manner but at doses that cannot be achieved eating natural products! Even this paper mentions this in the following statement: 

Lines 500-506: “Usually, a positive influence of polyphenols is observed for plant extracts or natural products (like berries, cocoa, grape polyphenols, red wine polyphenols, berries extract, tannin-rich products, pomegranate extract), that are composed of various polyphenols in their more “natural" form (e.g. , glycosides), while negative impact applies in particular to purified polyphenolic compounds, mainly aglycones [213,224], a form that is rarely found in food, but is a common form in dietary supplements.

Other statements that contradict the title: 

Lines 606-608: “There is also some "dangerous" plants, that if consumed along with medications permanently taken, should be carefully controlled, and the concomitant consumption should be under the supervision of a physician.

Lines 676-678: “it seems that in healthy people isoflavones do not pose a threat, but in the case of people with complete deficiencies or diseases, the situation may be different.

9. Prooxidant activity of polyphenols and the consequences 

Most of the cited studies used pure polyphenols that can be dangerous, however, no such molecules are found in real food.

interaction between metal ions, such as copper, and flavonoids might affect the outcome of therapeutic uses of the latter” (a paper cited by this manuscript), meaning flavonoids are beneficial.

Lines 767-769: ending the section with the following sentence “Therefore, further research should be intensified to explain under what conditions an antioxidant changes its face and begins to be a prooxidant.”, clearly the title of the section needs to be changed

10. Mutagenic, cancerogenic and genotoxic effects 

Again, the example were from in vitro studies, at doses that cannot be achieved eating natural food.

Other observations:

Lines 324-341: these two paragraphs can be deleted; they do not focus on polyphenols 

Lines 357-371: again, in this paragraph the topic is enzymes and not polyphenols

I advise the authors to adjust the paper based on  the idea that polyphenol supplementation could have possible side effects especially in vulnerable subpopulations.

Author Response

Our answers are in attached file.

Round 2

Reviewer 2 Report

Thank You for your answers and corrections of manuscript.

Reviewer 3 Report

The authors addressed the suggestions and significantly changed the manuscript that it is much improved.